# Exhaustive exercise abolishes REV-ERB-α circadian rhythm and shifts the kynurenine pathway to a neurotoxic profile in mice

Alisson Luiz da Rocha[1,2], Ana Paula Pinto[3] ⓘD, Ivo Vieira de Sousa Neto[3], Vitor Rosetto Muñoz[3] ⓘD, Bruno Brieda Marafon[2] ⓘD, Lilian Eslaine Costa Mendes da Silva[4] ⓘD, José Rodrigo Pauli[1] ⓘD, Dennys Esper Cintra[1] ⓘD, Eduardo Rochete Ropelle[1] ⓘD, Fernando Moreira Simabuco[5], Leandro Pereira de Moura[1], Ellen Cristini de Freitas[2] ⓘD, Donato Americo Rivas[6] and Adelino Sanchez Ramos da Silva[2,3] ⓘD

[1]*School of Applied Sciences, University of Campinas (UNICAMP), Limeira, São Paulo, Brazil*
[2]*Department of Health Sciences, Postgraduate Program in Rehabilitation and Functional Performance, Ribeirão Preto Medical School, University of São Paulo (USP), Ribeirão Preto, São Paulo, Brazil*
[3]*School of Physical Education and Sport of Ribeirao Preto, University of São Paulo (USP), Ribeirão Preto, São Paulo, Brazil*
[4]*Department of Ophthalmology, School of Medicine of Ribeirão Preto, University of São Paulo, Ribeirão Preto, São Paulo, Brazil*
[5]*Department of Biochemistry, Federal University of São Paulo (UNIFESP), São Paulo, São Paulo, Brazil*
[6]*Center for Exercise Medicine Research, Fralin Biomedical Research Institute, Virginia Tech Carilion, Roanoke, VA, USA*

Handling Editors: Karyn Hamilton & Mark Viggars

The peer review history is available in the Supporting Information section of this article (https://doi.org/10.1113/JP288290#support-information-section).

*The Journal of Physiology*

**Abstract figure legend** Acute exhaustive exercise (EE) reduces REV-ERB-α protein levels in skeletal muscle, which impairs its transcriptional repression of kynurenine (KYN) 3-monooxygenase (KMO), a key enzyme in the KYN pathway that converts KYN into potentially neurotoxic metabolites, collectively referred to as 'kynurenines'. This study identifies KMO as a novel transcriptional target of REV-ERB-α. Concurrently EE downregulates KYN aminotransferase

1 (KAT1), the principal enzyme that catalyses the conversion of KYN into kynurenic acid (KYNA). This neuroprotective and anti-inflammatory end product does not cross the blood–brain barrier. These molecular changes increase KYN availability and its influx into the brain, where KAT1 expression diminishes, favouring its metabolism towards neuro-toxic derivatives. Prolonged EE (i.e. overtraining) sustains the reduced capacity of skeletal muscle to metabolize KYN into KYNA due to persistently low KAT1 expression and is associated with reduced *Nr1d1* expression amplitude over 18 h in the hippocampus. These alterations may underlie neuronal dysfunction and contribute to behavioural and cognitive disturbances observed in overtraining syndrome.

**Abstract** The circadian-regulated transcriptional repressor REV-ERB-$\alpha$ is a key mediator of skeletal muscle oxidative capacity, enhancing exercise performance when activated. Conversely its global genetic ablation leads to impaired performance. Simultaneously the kynurenine (KYN) pathway, involved in tryptophan degradation, produces neurotoxic metabolites under stress and inflammation, contributing to CNS dysfunction and fatigue. These mechanisms may underlie the fatigue and performance impairments caused by exhaustive exercise (EE). This study investigated the interplay between REV-ERB-$\alpha$ and the KYN pathway in acute and chronic EE models. Time course analyses revealed that EE downregulated REV-ERB-$\alpha$ in skeletal muscle, correlated with KYN pathway alterations. Notably KYN metabolism shifted towards a neurotoxic profile, characterized by reduced KYN aminotransferase 1 (KAT1) and increased KYN 3-monooxygenase (KMO) expression in skeletal muscle, with increased KYN levels in the hippocampus. *In vitro* experiments using C2C12 myoblasts showed that REV-ERB-$\alpha$ knockout upregulated KAT1 and KMO, whereas overexpression selectively reduced KMO. Pharmacological activation of REV-ERB-$\alpha$ with SR9009 upregulated KAT1 in skeletal muscle and reduced KMO in the hippocampus of mice. These findings reveal a dynamic relationship between REV-ERB-$\alpha$ and the KYN pathway, linking peripheral and central responses to EE. This study highlights REV-ERB-$\alpha$ and the KYN pathway as critical regulators of exercise-induced fatigue and suggests potential therapeutic targets to mitigate its effects, offering novel insights into the molecular basis of performance impairments associated with EE.

(Received 2 December 2024; accepted after revision 4 June 2025; first published online 23 June 2025)

**Corresponding authors** A. L. da Rocha: School of Applied Sciences, University of Campinas (UNICAMP), Rua Pedro Zaccaria, 1300, Limeira, São Paulo 13484-350, Brazil. Email: alisson.rocha@alumni.usp.br

A. S. R. da Silva: School of Physical Education and Sport of Ribeirão Preto, University of São Paulo (USP), Avenida Bandeirantes, 3900, Monte Alegre, Ribeirão Preto, São Paulo 14040–907, Brazil. Email: adelinosanchez@usp.br

**Key points**

- Excessive exercise can impair performance and induce fatigue; however the underlying biological mechanisms remain incompletely understood.
- Although REV-ERB-$\alpha$ activation enhances skeletal muscle oxidative capacity and exercise performance, its deletion impairs both parameters.
- This study demonstrates that excessive exercise decreases REV-ERB-$\alpha$ levels in skeletal muscle and disrupts the kynurenine (KYN) pathway by downregulating KYN aminotransferase 1 (KAT1), an enzyme involved in a neuroprotective branch of the pathway.
- These alterations affect both skeletal muscle and the brain, suggesting a potential link between physical fatigue and brain function.
- REV-ERB-$\alpha$ suppresses KYN 3-monooxygenase (KMO), a key enzyme in the KYN pathway that promotes the formation of potentially neurotoxic metabolites, thereby revealing a novel mechanism and a potential therapeutic target.

## Introduction

Although regular moderate-intensity exercise provides non-pharmacological benefits for managing several diseases (Pedersen & Saltin, 2006), prolonged exhaustive exercise (EE) is associated with various peripheral (da Rocha et al., 2017, 2018; Pereira et al., 2016) and central

maladaptations (Pinto et al., 2017), suggesting a potential molecular dysregulation contributing to EE-induced performance decline. Notably EE is not exclusive to elite athletes but also occurs in the general population, often manifesting as a behavioural addiction associated with cognitive and metabolic disturbances (Chhabra et al., 2024). Although the molecular mechanisms underlying chronic EE-induced metabolic impairment are not yet fully understood, Woldt et al. (2013) demonstrated that global knockout (KO) of REV-ERB-α, encoded by the nuclear receptor subfamily 1, group D, member 1 (*Nr1d1*) gene, reduces exercise capacity and maximal oxygen consumption ($VO_2$max) in mice.

One prominent effect of EE is chronic low-grade inflammation, often characterized by elevated interleukin-6 (IL-6) levels in various tissues (da Rocha et al., 2019). IL-6 and other proinflammatory cytokines activate indoleamine 2,3-dioxygenase 1 (IDO1), whereas stress-related signals may activate tryptophan dioxygenase 2 (TDO2), promoting tryptophan (TRP) degradation via the kynurenine (KYN) pathway (Schwarcz et al., 2012). This leads to the production of L- KYN, which can be converted into KYN acid (KYNA) by KYN aminotransferases (KATs) or into 3-hydroxy-KYN (3-HK) by KYN 3-monooxygenase (KMO). 3-HK can then be further metabolized into picolinic acid (PIC) or quinolinic acid (QUIN), the latter of which is converted to nicotinamide (NAM), an essential $NAD^+$ precursor (Savitz, 2020). Inflammation skews this pathway towards the KMO branch, generating neuro-active metabolites contributing to neuroinflammation and neurodegeneration (Marx et al., 2021; Mor et al., 2021). Several regions of the CNS may be affected by these neuroactive molecules; however the hippocampus appears to be the most significantly influenced in the context of physical exercise (Hamilton & Rhodes, 2016). Inflammation-induced impairment of hippocampal neurogenesis can contribute to cognitive and behavioural disturbances.

Moravcová et al. (2022) demonstrated that circadian clock genes regulate KYN pathway enzymes and that lipopolysaccharide (LPS)-induced proinflammatory response disrupts the circadian expression of *Nr1d1* and KYN enzymes in the pineal gland, heart and liver of rats. REV-ERB-α regulates autophagy in skeletal muscle and selectively modulates proinflammatory cytokines in macrophages of mice (Sato et al., 2014; Woldt et al., 2013). These findings suggest that EE may suppress *Nr1d1*, shifting KYN pathway activity towards a neuro-toxic profile and linking it to low-grade inflammation. Analysis using the MetaMEx database confirmed that acute exercise reduces *NR1D1* expression in human skeletal muscle within 2–6 h, highlighting significant circadian shifts that may influence muscle adaptations, as suggested by da Rocha et al. (2022).

Therefore after an acute and chronic EE we conducted time course analyses of REV-ERB-α and KYN pathway mRNA and protein levels in skeletal muscle and hippocampus. KYN metabolites were also evaluated in these tissues and serum. After an 8-week overtraining (OT) protocol we identified strong correlations between REV-ERB-α and KMO, prompting *in vitro* experiments to assess KYN pathway enzyme responses to REV-ERB-α KO, overexpression and pharmacological modulation. Finally we validated these findings *in vivo* using the REV-ERB-α agonist SR9009, assessing responses in skeletal muscle and hippocampus.

## Methods

### Experimental animals

Eight-week-old male C57BL/6 mice from the Central Animal Facility of the Ribeirão Preto campus from the University of São Paulo (USP) were used. In this study only male mice were used to eliminate potential confounding effects of the oestrous cycle on data inter-pretation. Additionally 8-week-old mice correspond to the stage of sexual maturity (Dutta & Sengupta, 2016), which is analogous to the phase of specialization in elite athletes (De Bosscher et al., 2023). The animals were accommodated in sterile micro-insulators (three animals per cage) in a ventilated rack (INSIGHT, Ribeirão Preto, SP, Brazil) with a controlled temperature ($22 \pm 2$°C) on a 12:12-h light–dark cycle (light: 6.00 a.m. to 6.00 p.m.; dark: 6.00 p.m. to 6.00 a.m.). Food

**Alisson L. da Rocha** (left) holds a PhD in sciences from the Ribeirao Preto Medical School – University of São Paulo, where he conducted research in Dr. Adelino laboratory. He is currently a postdoctoral fellow at the University of Campinas and the University of Michigan, supervised by Dr. Jose Pauli and Dr. Jorge Ruas, respectively. His research focuses on physiological and molecular responses to overtraining, particularly the KYN pathway in muscle–brain communication. **Adelino Sanchez Ramos da Silva** (right) is an associate professor at the University of São Paulo – School of Physical Education and Sport of Ribeirão Preto. His research investigates molecular mechanisms underlying adaptations to physical exercise. He has extensive experience analysing exercise-induced signalling in various tissues under physiological and pathological conditions. His work emphasizes chronic degenerative diseases, such as type 2 diabetes, obesity and sarcopenia, to understand how exercise modulates molecular responses in health and disease.

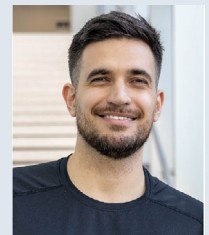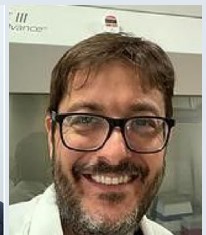

(Purina chow; 63.4% carbohydrate, 25.9% protein, and 10.6% lipids) and water were provided *ad libitum*. All experimental procedures were performed according to the Brazilian College of Animal Experimentation (COBEA). Also the experimental procedures were approved by the Ethics Committee of the University of São Paulo (I.D. 2017.5.33.90.7). Figure 1 shows the graphical representations of the experimental designs.

These mice were randomly distributed and used in three different experiments. In experiment 1 mice were distributed into two experimental groups: control (CTL; the group was not subjected to any experimental manipulation; $n = 20$) and exhaustive exercise (EXH; the group was submitted to an acute exhaustive endurance exercise protocol; $n = 20$). In experiment 2 mice were distributed into two experimental groups: sedentary (SED; $n = 20$) and overtrained (OTR; the group was submitted to a chronic exhaustive endurance exercise; $n = 20$). The

animals in these two experimental designs were submitted to 5 days of adaptation on a treadmill (INSIGHT, Ribeirão Preto, SP, Brazil), 10 min per day, at a 6 m/min speed. In experiment 4 mice were distributed into two experimental groups: vehicle (VEH; the group was treated via I.P. with vehicle solution containing cremophor 30% diluted into ultrapure water; $n = 6$) and SR9009 (the mice were treated twice a day via I.P. with SR9009, a REV-ERB-$\alpha$ agonist; $n = 6$). The sample size for each group/time point is described in the figure legends.

### *In vitro* experiments

For experiment 3 C2C12 myoblasts (ATCC CRL-1772) were cultured in Dulbecco's modified Eagle's medium (DMEM, #12,100,046, Thermo Fisher scientific) +10% of fetal bovine serum (FBS – Gibco #A476680) + 1% of penicillin/streptomycin complexes (P/S – Gibco #15,140,122) at 37°C, 5% $CO_2$.

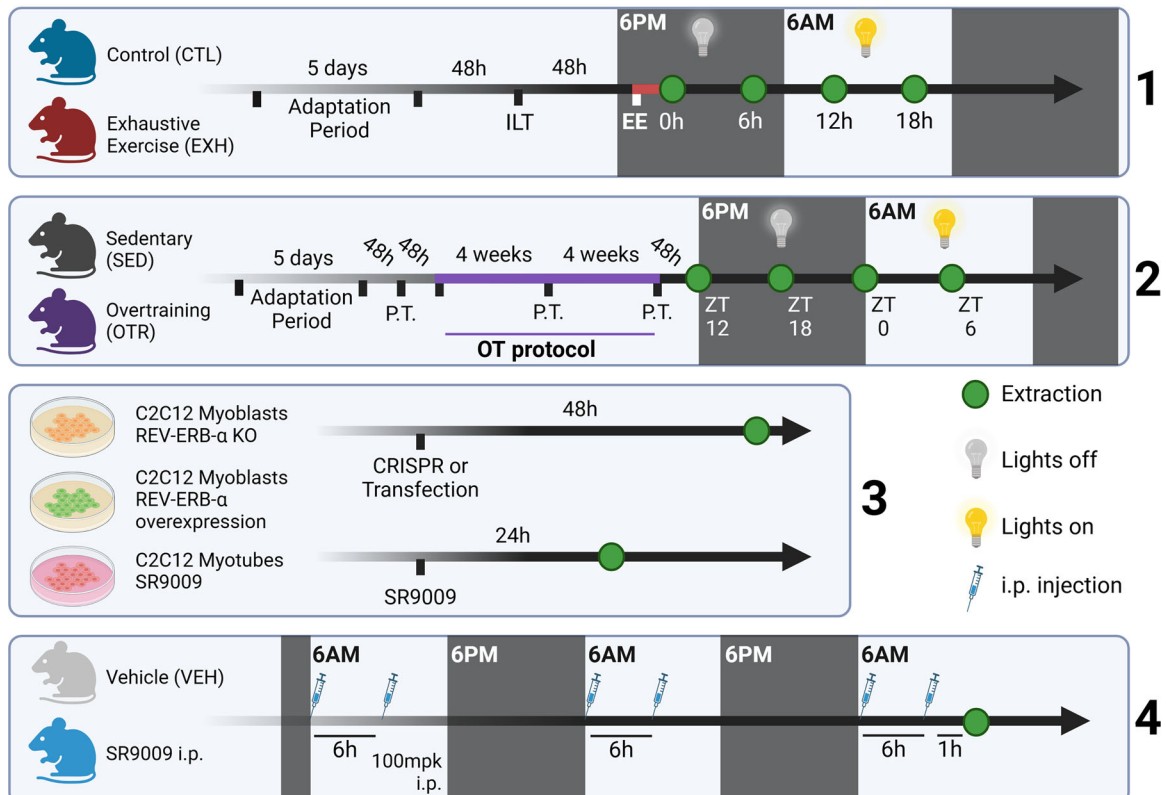

**Figure 1. Graphical representation of the experimental designs**
In experiment 1 mice were divided into two groups: CTL (control) – sedentary mice that did not undergo any experimental manipulation, and EXH (exhaustive exercise) – mice that underwent a week-long adaptation period on a treadmill, performed the incremental load test (ILT), and participated in an acute exhaustive exercise (EE) running session. In experiment 2 mice were divided into two groups: SED (sedentary) – sedentary mice that did not undergo any experimental manipulation, and OTR (chronic exhaustive endurance exercise) – mice that underwent a week-long adaptation on a treadmill, performed physical tests (PT) and followed an 8-week overtraining (OT) protocol. In experiment 3 C2C12 cells with REV-ERB-$\alpha$ knockout, overexpression or agonism (SR9009) were utilized to analyse proteins related to the kynurenine (KYN) pathway. In experiment 4 mice received two daily doses of SR9009 or vehicle solution (30% cremophor in ultrapure water) for 3 days. Created in BioRender. Da Rocha, A. (2025). [Colour figure can be viewed at wileyonlinelibrary.com]

To generate REV-ERB-α knockout cells a single-guide RNA (sgRNA) for the exon 4 of REV-ERB-α (sequence: 5′GTTGCGATTGATGCGAACGATGG 3′) was cloned into the PX459 vector (SpCas9(BB)-2A-Puro V2.0, Addgene, #62,988). Cells were transfected using Lipofectamine CRISPRMAX (Thermo Scientific, #CMAX00001) and selected using puromycin (1 µg/mL for 48 h). Selected cells were separated into single cells in a 96-well plate. The single colonies were grown and genotyped to obtain homozygous deletion for REV-ERB-α. For REV-ERB-α overexpression C2C12 cells were seeded in a 6-well plate at 50% confluence and transfected with pcDNA 3.1-GFP or pcDNA-REV-ERB-α using Lipofectamine 3000 (Thermo Scientific, #L3000150). After 48h of transfection the cells were washed with $1\times$ phosphate-buffered saline (PBS) and lysed with RIPA buffer for immunoblotting analysis. Another set of C2C12 cells was treated with the REV-ERB-α agonist (SR9009, 5 µM, #11,929 from Cayman) for 24 h and subsequently processed for immunoblotting.

### Incremental load test

For experiment 1 mice were submitted to the incremental load test (ILT) 48 h after the treadmill adaptation. The ILT started at an initial velocity of 6 m/min, at −14% inclination (i.e. downhill running), with 3 m/min increments every 3 min until voluntary exhaustion. The maximum velocity reached was classified as maximum aerobic power (MAP) intensity. This value was used to prescribe the acute exhaustive exercise intensity for the EE group. For experiment 2 mice were submitted to the ILT test 48 h after the adaptation period and 48 h after the last training session at weeks 4 and 8. The ILT was performed as described previously; however the first evaluation was performed at 0% inclination and the final two at −14%. The values obtained were utilized to prescribe exercise intensity through the 8-week training protocol. Downhill running, characterized by predominant eccentric muscle contractions, was incorporated into both protocols to maximize muscle damage during the exercise/training phase (Proske & Allen, 2005).

### Rotarod test

The rotarod test, a widely used method for assessing motor coordination and balance (Brooks et al., 2012), was employed as a supplementary evaluation to monitor performance throughout the chronic exhaustive exercise protocol. Only the animals from experiment 2 performed this test, which was realized 24 h after the ILT at weeks 0, 4 and 8. The apparatus was programmed to start at an initial speed of 1 rpm and reach a final speed of 40 rpm, which was achieved 300 s after the initiation of the movement. The acceleration throughout the entire test was constant. Three attempts were made per animal, with a 10 s interval between each attempt, and the time (in seconds) until the fall of each animal was recorded.

### Acute exhaustive exercise protocol

After 48 h of the ILT test mice were weighed, and the exercise session was initiated between 6.00 and 7.00 p.m. The animals ran at 60% of the velocity obtained in ILT until exhaustion (characterized by the mouse's inability to run without touching the end of the treadmill, even with gentle stimulation with tweezers) at −14% inclination.

### Chronic exhaustive exercise protocol (overtraining protocol; OT)

Forty-eight hours after the rotarod test the OT protocol was initiated. The details of the protocol are presented in Table 1.

### Blood glucose dosage

The blood from the tail tip was collected, and glucose levels were measured before and immediately after the acute EE protocol using a glycaemic monitoring system (Accu-ChekTM Active model, Roche, Santo André, SP, Brazil).

### SR9009 treatment *in vivo*

The SR9009 powder (DC9544; DC Chemicals, Shangai, China) was diluted into a vehicle solution containing Kolliphor EL (Sigma, Missouri, MO, USA) 30% in ultrapure water. Mice received 100 mpk I.P. at 6.00 a.m. (the beginning of the light cycle) followed by a second injection 6 h later. The treatment protocol lasted three consecutive days. This protocol was based on a previous study, wherein SR9009 treatment efficiently countered the harmful effects of dexamethasone co-treatment (Mayeuf-Louchart et al., 2017).

### Tissue extraction

For experiment 1 the tissue extraction was realized immediately, 6, 12 and 18 h after the acute EE protocol. For experiment 2 the tissue extraction was initiated 48 h after the physical tests, at the beginning of the dark cycle (ZT12), and the extractions were followed every 6 h, completing four time points. For experiment 4 the tissue extraction was performed 1 h after the last injection. In all experiments mice were anaesthetized

**Table 1. Details of the chronic exhaustive exercise protocol (overtraining protocol)**

| Week | Intensity (%MAP) | Volume (min) | Daily sessions | Inclination (%) | Recovery between sessions (h) |
|---|---|---|---|---|---|
| 1 | 60 | 15 | 1 | 0 | 24 |
| 2 | 60 | 30 | 1 | 0 | 24 |
| 3 | 60 | 45 | 1 | 0 | 24 |
| 4 | 60 | 60 | 1 | 0 | 24 |
| 5 | 60 | 60 | 1 | −14 | 24 |
| 6 | 70 | 60 | 1 | −14 | 24 |
| 7 | 75 | 75 | 1 | −14 | 24 |
| 8 | 75 | 75 | 2 | −14 | 4 |

Abbreviation: MAP, maximum aerobic power.

by an I.P. administration of xylazine (10 mg/kg of body weight) and ketamine (100 mg/kg of body weight). As soon as the loss of pedal reflexes confirmed the effect of anaesthesia, mice were decapitated, and the blood was collected and subsequently centrifuged (3500 rpm, 15 min and 4°C) to obtain serum. Later the gastrocnemius and hippocampus samples were removed, washed with sterile saline and quickly frozen in liquid nitrogen. Afterwards the samples were stored at −80°C for further analysis using RT-quantitative PCR (RT-qPCR), immunoblotting and enzyme-linked immunosorbent assay (ELISA) techniques.

### RT-qPCR

Total RNA from skeletal muscles and hippocampus was extracted using TRIZOL (Invitrogen, Carlsbad, CA, USA). All procedures were performed under standard RNase-free conditions to avoid exogenous RNase contamination. The real-time qPCR technique was conducted using the StepOne Real-Time PCR System (Life Technologies Corporation) for the analysis of mRNA expression for *Nr1d1* (forward 3'-5' AGAGAGGCCATCACAACCTC, reverse 3'-5' TGTAGGTGATAACACCACCTGT), *Kyat1* (forward 3'-5' AAAGGAACAGACTTCTGCAACC, reverse 3'-5' CCACACACAGTTCTGCTTCAG) and *Kmo* (forward 3'-5' CAAGGAATGAATGCGGGCTT, reverse 3'-5' ATGCGCTCGCATCTCTATGT).

RT-qPCR was performed using the following reagents: 5 μL of SYBR Green Master Mix (Bio-Rad, California, CA, USA), 1 μL of forward primer, 1 μL of reverse primer (both at a final concentration of 100–200 nM), 1 μL cDNA (10 ng) and 2 μL of DEPC treated $H_2O$. Each amplification reaction occurred with standard cycling with the following cycles: one cycle at 95°C for 30 s, 40 cycles of 15 s at 95°C and 1 min at 60°C. Relative quantitation was calculated by the $2^{-\Delta\Delta CT}$ method using Thermo Fisher Cloud Software, RQ version 3.7 (Life Technologies Corporation). All values were corrected by the value obtained in *glyceraldehyde-3-phosphate dehydrogenase* (*Gapdh*; forward 3'-5' AGGTCGGTGTGAACGGATTTG, reverse 3'-5' TGTAGACCATGTAGTTGAGGTCA) amplification.

### Immunoblotting

The immunoblotting technique was performed as previously described by our research group (da Rocha et al., 2022). The antibodies used were REV-ERB-α (sc-393 215), KAT1 (sc-271 709) and histone H3 (sc-517 576) from Santa Cruz Biotechnology (Santa Cruz, CA, USA); p-REV-ERB-α Ser55/59 (#2129) and GAPDH (#2118) from Cell Signalling Technology (Cell Signalling Technology, MA, USA). KMO (10 698-1-AP) from Proteintech (Proteintech Group, IL, USA). All the primary antibodies were utilized at a dilution of 1:1000, and the secondary antibodies at a dilution between 1:5000 and 1:10,000. Images were acquired using the ChemiDoc Imaging System (Bio-Rad, CA, USA) and quantified using the software Image Studio for C-DiGit Blot Scanner. The subcellular fractionation was performed as described by Dimauro et al. (2012). All membranes were checked for data integrity using the Proofig pipeline (https://www.proofig.com).

### ELISA

The hippocampus and gastrocnemius samples were homogenized in PBS buffer and centrifuged at 5000*g* for 5 min at 4°C. The supernatant was collected and subsequently analysed for the measurement of total protein concentration using the BCA method (He, 2011). Equal amounts of protein and serum were used among the samples for subsequent analyses. The

following kits were used: MOFI01315 (Kynurenine) from Assay Genie (Dublin, Ireland); MBS7256170 (Kynurenic Acid) and MBS1607987 (Tryptophan) from Mybiosource (CA, USA). All analyses followed the protocols and recommendations provided by the manufacturer.

### Bioinformatic analysis

The impact of acute aerobic, high-intensity interval training (HIIT) and exercise time course effect on *NR1D1* gene expression in human skeletal muscle was evaluated using the MetaMEx database. Original studies used for the meta-analysis are publicly available on the GEO repository (https://www.ncbi.nlm.nih.gov/geo). The curated database (MetaMEx) generated during the current study is available at www.metamex.eu. For each study $n$ size, logFC and adjusted $P$-value were used to fit a random effect model to the data. The bottom line of each graph shows the meta-analysis score. The $P$-values are adjusted for multiple comparisons using the Bonferroni method. In addition we explored the effects of exercise on *NR1D1*, *KMO* and *KYAT1* gene expression in mono-nuclear cells (GSE137832 database; $n = 5$, healthy male). Blood samples for gene expression measurements were analysed 1 h after exercise.

### Statistical analysis

Results are expressed as mean $\pm$ SD. Levene's test was used to verify the homogeneity of variances, and the Shapiro–Wilk test was used to check data normality. When normality was confirmed, a two-way ANOVA was used to compare the response of a specific protein/gene expression between exercise time and groups. Tukey's post hoc test was performed when the two-way ANOVA indicated significance. An unpaired Student's $t$ test was applied to investigate the possible differences between the two experimental groups. The paired Student's $t$ test was used to compare the blood glucose levels before and after the EE session. Pearson's or Spearman's correlations were utilized to investigate possible correlations between REV-ERB-$\alpha$, *Nr1d1* and KYN pathway enzymes in skeletal muscle and hippocampus. All statistical analyses were set at $P \leq 0.05$ and two-sided. Statistical analyses were performed using GraphPad Prism version 8.0.1 for Windows (GraphPad Software, CA, USA). Rhythmicity and amplitude analyses were performed using the MetaCycle package (Wu et al., 2016) in R (version 4.1.3; R Core Team, Vienna, Austria). Specifically the JTK_CYCLE algorithm was applied to assess diurnal oscillations in gene and protein expression over an 18 h time course based on samples collected at defined time points (Hughes et al., 2010).

## Results

### Acute EE decreased blood glucose levels immediately after the exercise session, whereas chronic EE impaired physical performance accompanied by a reduction in body weight

The groups did not present differences in body weight for the acute EE session (Fig. 2*A*). The EXH group showed a 45% decrease in blood glucose immediately after the exercise session (Fig. 2*B*). The OTR group exhibited a reduction in body weight after the 8-week protocol compared to SED (Fig. 2*E*). Also this group presented an increase in maximum aerobic power after the first 4 weeks of the OT protocol and a decrease at week 8 (Fig. 2*F*).

### REV-ERB-$\alpha$ responses to acute exhaustive exercise

**Hippocampus: EE increased the REV-ERB-$\alpha$ phospho/total ratio after the first 6 h, with a reduction in total levels observed at the 18 h time point.** The EXH group reduced the total REV-ERB-$\alpha$ protein levels compared to the CTL group at 18 h in the hippocampus ($P = 0.007$; Fig. 3*A*). The same group showed an upregulation of $^P$REV-ERB-$\alpha^{(Ser55/59)}$ immediately after the exercise session (around 5 times) compared to CTL ($P = 0.023$; Fig. 3*B*). The EXH group exhibited a higher pREV-ERB-$\alpha^{(Ser55/59)}$ / REV-ERB-$\alpha$ ratio at 0h ($P = 0.006$) and 6 h ($P = 0.038$) compared to CTL (Fig. 3*C*). Also this group showed higher mRNA levels of *Nr1d1* at 18 h compared to CTL ($P = 0.042$; Fig. 3*D*).

**Skeletal muscle: EE extinguished the diurnal rhythm of REV-ERB-$\alpha$ accompanied by an increase in the phospho/total ratio after the first 6 h.** The EXH group reduced the protein levels of total REV-ERB-$\alpha$ in the gastrocnemius immediately ($P < 0.0001$) and 6 h ($P = 0.008$) after the acute EE session compared to the CTL group (Fig. 3*E*). There were no differences between groups for $^P$REV-ERB-$\alpha^{(Ser55/59)}$ protein levels and *Nr1d1* gene expression (Fig. 3*F* and *H*). The EXH group exhibited a higher ratio between $^P$REV-ERB-$\alpha^{(Ser55/59)}$ and total REV-ERB-$\alpha$ immediately ($P = 0.015$) and 6 h ($P = 0.049$) after the acute exhaustive session than CTL (Fig. 3*G*). Compared to the CTL group the EXH group displayed a reduced amplitude of REV-ERB-$\alpha$ expression in the gastrocnemius muscle over the 18 h postexercise period ($P = 0.001$; Fig. 3*I*).

**Skeletal muscle subcellular fractionation: reduced REV-ERB-$\alpha$ content in nuclear and cytosolic compartments immediately after exhaustive exercise.** Subcellular fractionation is an indispensable method in molecular biology, facilitating the examination of cellular mechanisms. By meticulously isolating and analysing

distinct cellular compartments like the nucleus and cytoplasm, it provides further information on molecule localization, function and regulation across various subcellular domains. The EXH group exhibited a reduction in total REV-ERB-$\alpha$ protein content in nuclear ($P = 0.006$) and cytosolic ($P = 0.044$) compartments compared to CTL immediately after the exhaustive exercise session (Fig. 3*K* and *L*).

**Publicly available transcriptomic datasets of humans.** The meta-analysis (MetaMEx) included data from 46 studies of acute aerobic exercise with 362 individuals. There was a marginal trend towards significance (LogFC = $-0.68$; $P = 0.0960$) for a lower *NR1D1* gene expression in skeletal muscle after acute aerobic exercise (Fig. 3*M*). Moreover regardless of modality, intensity and skeletal muscle analysed exercise-induced changes in *NR1D1* expression were lowest in studies where skeletal muscle biopsies were taken after a recovery period (2–6 h) compared to the baseline (Fig. 3*N*), suggesting a significant reduction in these periods.

**Total REV-ERB-$\alpha$ displays an inverse relationship between phosphorylation status and gene expression in skeletal muscle at baseline.** At baseline in the gastrocnemius muscle REV-ERB-$\alpha$ exhibited an inverse correlation between its total and phosphorylated (Ser55/59) protein levels ($P = 0.01$; $r = -0.53$), as well as between

its total protein and *Nr1d1* mRNA levels ($P = 0.05$; $r = -0.46$). Additionally phosphorylated REV-ERB-$\alpha$ (Ser55/59) positively correlated with *Nr1d1* mRNA levels ($P = 0.01$; $r = 0.59$), as illustrated in Fig. 4*A* and *B*.

### Effects of acute EE protocol on the enzymes and metabolites of the kynurenine pathway

**Hippocampus: EE immediately reduced KAT1 levels, accompanied by an increase in kynurenine bioavailability, followed by a delayed suppression of KMO gene expression.** The EXH group reduced the hippocampus KAT1 protein levels immediately after the exercise session compared to the CTL group ($P = 0.044$; Fig. 5*A*). There were no differences between groups in KMO protein levels (Fig. 5*B*) and *Kyat1* mRNA levels (Fig. 5*C*). The EXH downregulated the *Kmo* mRNA levels at 18 h compared to CTL ($P = 0.010$; Fig. 5*D*). The EXH group increased KYN bioavailability in the hippocampus immediately after the exercise session ($P = 0.028$; Fig. 5*I*; KYN/TRP ratio).

**Skeletal muscle: EE reduced KAT1 at both the protein and transcriptional levels, accompanied by reduced KYN and higher KYNA bioavailability immediately after, with a delayed upregulation of KMO gene expression.** The EXH group reduced the KAT1 protein levels in the gastrocnemius at 0 h ($P < 0.0001$) and 6 h ($P = 0.001$) compared to the CTL group (Fig. 5*E*). There were

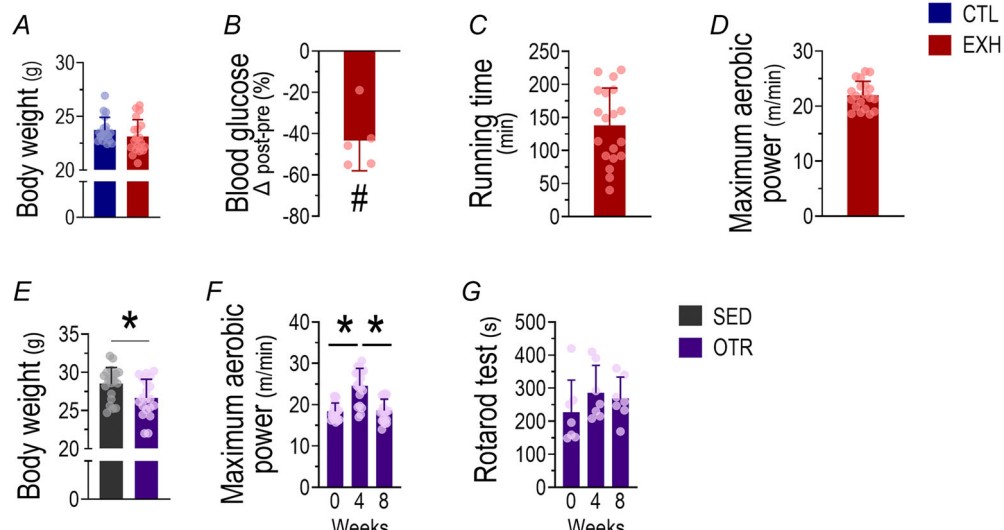

**Figure 2. Physiological and performance outcomes following acute and chronic exhaustive exercise protocols**
Mean ± standard deviation of body weight in CTL and EXH groups (*A*). Changes in blood glucose before and after the exhaustive exercise protocol (*B*). Running time in the acute exhaustive exercise test (*C*). Maximum aerobic power of the EXH group (*D*). Body weight of SED and OTR groups at week 8 (*E*). Maximum aerobic power in the incremental load test for the OTR group at weeks 0, 4 and 8 (*F*). Time spent on the rotarod test in the OTR group (*G*). $n = 18$–20 for *A* and *C*–*F*; $n = 5$ for *B*; $n = 7$ for *G*. #$P \leq 0.05$ *vs.* baseline; *$P \leq 0.05$. CTL, control group; EXH, acute exhaustive exercise group; SED, sedentary group; OTR, chronic exhaustive exercise group. [Colour figure can be viewed at wileyonlinelibrary.com]

no differences between groups in KMO protein levels (Fig. 5F). The EXH group downregulated the *Kyat1* mRNA levels compared to CTL at 0 h ($P = 0.0001$), 6 h ($P < 0.0001$) and 12 h ($P = 0.025$; Fig. 5G). On the contrary this group upregulated the *Kmo* mRNA levels at 18 h compared to CTL ($P = 0.048$; Fig. 5H). The EXH group reduced KYN/TRP ($P = 0.048$) and increased KYNA/TRP ($P = 0.005$) ratios in the gastro-

cnemius immediately after the exercise session (Fig. 5I and J). Also this group showed a reduced ratio between the KYN/KYNA ratio ($P = 0.008$; Fig. 5K).

**Serum: EE reduced the KYNA levels immediately after the EE.** The EXH group showed decreasing levels of KYNA/TRP in serum compared to the CTL group ($P = 0.031$; Fig. 5J). Additionally the ratio of KYN/KYNA

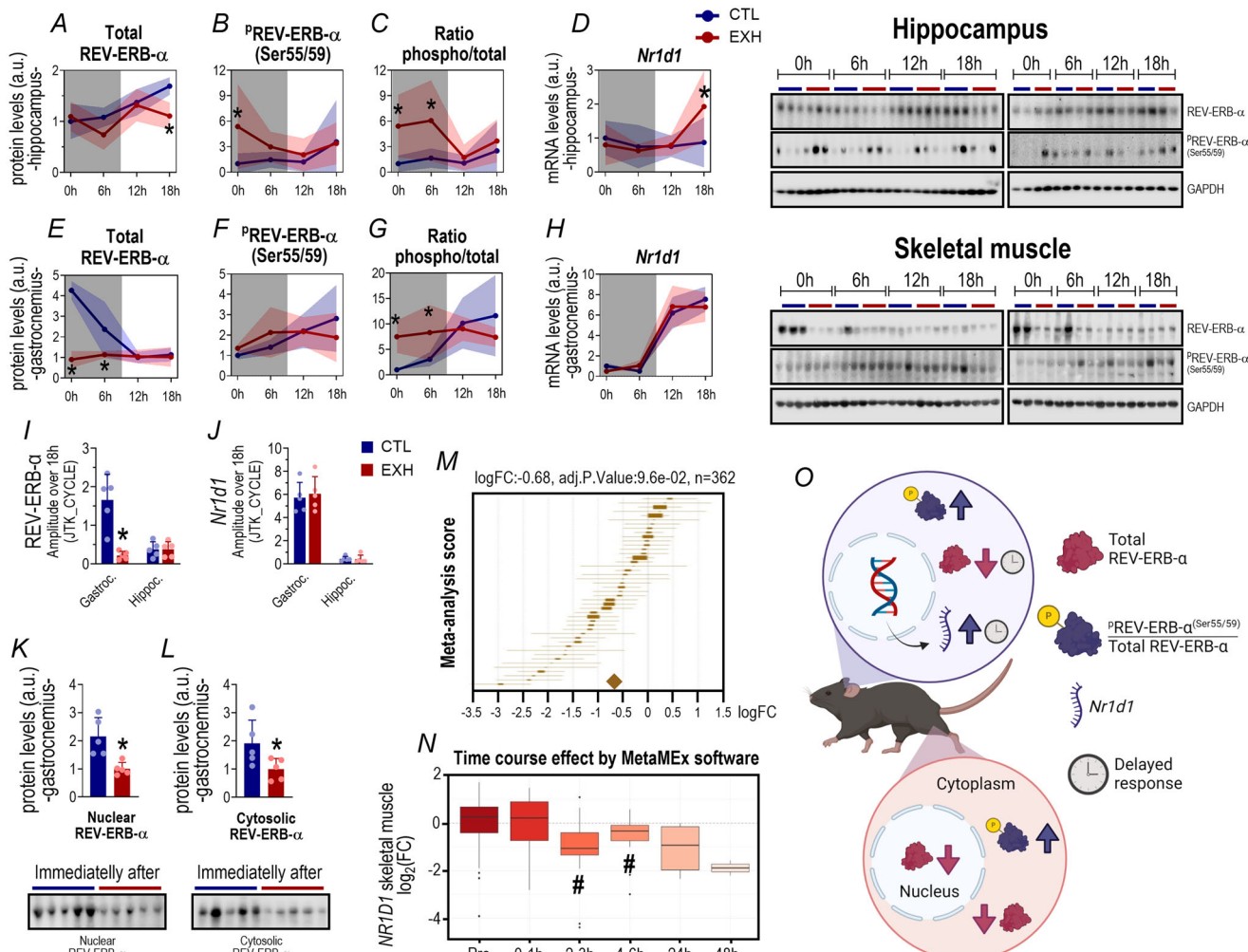

**Figure 3. REV-ERB-α and *Nr1d1* responses to acute exhaustive exercise in hippocampus and skeletal muscle**

Comparisons between groups for total REV-ERB-α (A), ᵖREV-ERB-α (Ser55/59) (B), the ratio between phosphorylated/total REV-ERB-α (C) protein levels and *Nr1d1* mRNA levels (D) in the hippocampus. Comparisons between groups for total REV-ERB-α (E), ᵖREV-ERB-α (Ser55/59) (F), the ratio between phosphorylated/total REV-ERB-α (G) protein levels and *Nr1d1* mRNA levels (H) in the gastrocnemius. Amplitude analysis over 18 h through JTK_CYCLE for REV-ERB-α (I) and *Nr1d1* (J) in the gastrocnemius and hippocampus. Nuclear (K) and cytosolic (L) REV-ERB-α protein levels in the gastrocnemius immediately after the acute exhaustive exercise session. The online tool MetaMEx (www.metamex.eu) allows for the quick interrogation of all published exercise studies for a single gene. The forest plot of individual statistics (fold-change, FDR, 95% confidence intervals) and meta-analysis score in acute aerobic exercise (M). Multiple comparisons show the time course effect in human skeletal muscle, indicating that acute exercise, regardless of modality, reduced *NR1D1* gene expression after 2–6 h compared to the baseline (N). Summary of the present data, highlighting the major responses of REV-ERB-α to an acute exhaustive exercise stimulus (O). Data correspond to the mean ± SD of $n = 4–5$ mice. *$P \leq 0.05$. CTL, control group; EXH, acute exhaustive exercise group. [Colour figure can be viewed at wileyonlinelibrary.com]

tended to be higher ($P = 0.060$; Fig. 5*K*), suggesting a potential neurotoxic imbalance.

### REV-ERB-α adaptations to overtraining

To analyse the pattern related to the circadian cycle, the Zeitgeber time (ZT) was adopted: ZT0 – beginning of the light phase and ZT12 – beginning of the dark phase. This strategy differed from that used in the acute exercise group, as the time course began only after the completion of the acute exercise session.

**Hippocampus: OT reduced the phospho/total ratio at ZT12, with an increase in total REV-ERB-α protein levels at ZT6.** The OTR group presented higher REV-ERB-α protein levels when compared to SED at ZT6 ($P = 0.042$; Fig. 6*A*). The groups did not show differences in $^{P}$REV-ERB-α$^{(Ser55/59)}$ protein levels (Fig. 6*B*). The OTR group exhibited a lower $^{P}$REV-ERB-α$^{(Ser55/59)}$/total REV-ERB-α ratio at ZT12 compared to SED ($P = 0.037$; Fig. 6*C*). There were no differences between *Nr1d1* levels between groups (Fig. 6*D*).

**Skeletal muscle: at ZT6, OT reduced total REV-ERB-α protein levels while increasing the phospho/total ratio, accompanied by a concomitant increase in *NR1D1* gene expression.** The OTR group presented lower total REV-ERB-α protein levels than the SED group at ZT6 ($P = 0.05$; Fig. 6*I*). The groups did not show differences in $^{P}$REV-ERB-α$^{(Ser55/59)}$ protein levels (Fig. 6*J*). However the OTR group exhibited higher values in $^{P}$REV-ERB-α$^{(Ser55/59)}$/total REV-ERB-α ratio ($P = 0.0003$) and *Nr1d1* mRNA ($P = 0.007$) levels at ZT6 (Fig. 6*K* and *L*, respectively).

### Effects of overtraining on enzymes and metabolites of the kynurenine pathway

**Hippocampus: OT downregulated both KYAT1 and KMO gene expressions at ZT12 and ZT18.** There were no differences between groups in KAT1 and KMO protein levels (Fig. 6*E* and *F*). The OTR group downregulated the *Kyat1* ($P < 0.0001$ and $P = 0.0006$) and *Kmo* ($P = 0.049$ and $P = 0.002$) mRNA levels at ZT12 and ZT18 compared to SED (Fig. 6*G* and *H*).

**Skeletal muscle: OT reduced KAT1 expression along the diurnal rhythm and increased KMO gene expression at ZT6.** The OTR group reduced the KAT1 protein levels in gastrocnemius, showing lower values at ZT0 ($P = 0.011$), ZT6 ($P = 0.015$) and ZT18 ($P = 0.034$) than SED (Fig. 6*M*). There were no differences between groups in KMO protein levels (Fig. 6*N*). The OTR group down-regulated *Kyat1* ($P = 0.044$) and upregulated the *Kmo* ($P = 0.004$) mRNA levels at ZT6 compared to SED (Fig. 6*O* and *P*).

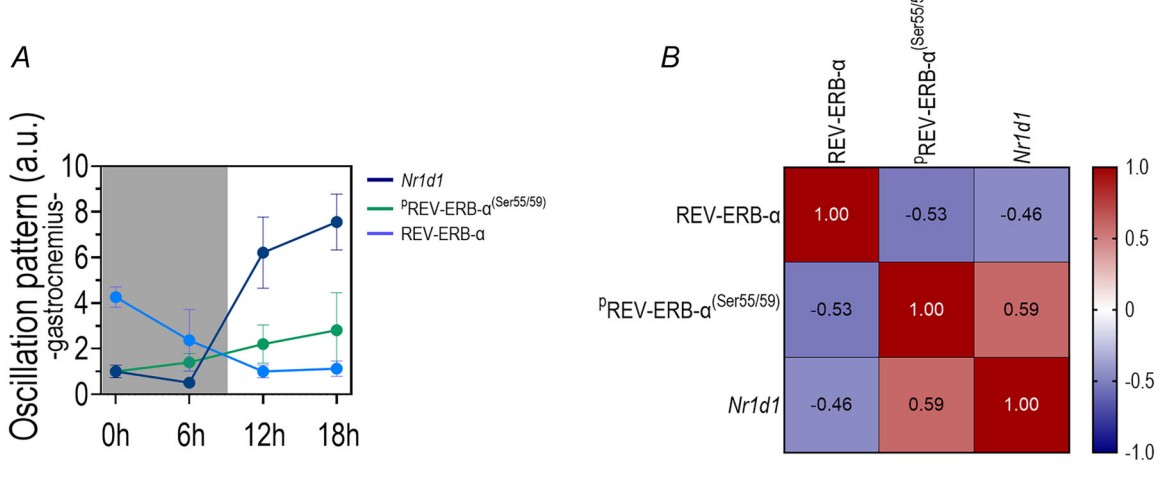

**Mice skeletal muscle at baseline**
**(CTL group)**

**Figure 4. Baseline expression and correlation of total and phosphorylated REV-ERB-α protein and *Nr1d1* mRNA in gastrocnemius muscle**
Expression patterns of total and phosphorylated REV-ERB-α protein and *Nr1d1* mRNA (*A*) and their correlations (*B*) at baseline in the gastrocnemius muscle of the control group. Data correspond to the mean ± SD; *n* = 18–20, CTL: control group. [Colour figure can be viewed at wileyonlinelibrary.com]

**Serum: OT did not alter kynurenine metabolite concentrations at ZT12.** No differences existed between groups for KYN/TRP, KYNA/TRP and KYN/KYNA ratios in serum at ZT12 (Fig. 6Q).

**NR1D1 shows reduced amplitude over 18 h in the hippocampus.** The group submitted to the OT protocol exhibited lower *Nr1d1* amplitude over 18 h in the hippocampus compared to the SED group ($P = 0.02$; Fig. 6S).

### REV-ERB-α exhibits a correlation with KMO levels in skeletal muscle and hippocampus following acute and chronic EE, as confirmed by bioinformatic analyses

The EXH group exhibited a negative correlation between REV-ERB-α and KMO protein levels (Fig. 7A; $r = -0.45$; $P = 0.046$) with an opposite pattern between *Nr1d1* and *Kmo* mRNA levels (Fig. 7B; $r = 0.76$; $P = 0.0006$) in gastrocnemius samples. Furthermore the chronic exhaustive exercise group (OTR) demonstrated a positive correlation between *Nr1d1* and *Kmo* mRNA levels in gastrocnemius

(Fig. 7C; $r = 0.44$; $P = 0.06$) and hippocampus (Fig. 7D; $r = 0.61$; $P = 0.004$) Additionally we supported our findings by utilizing compiled data from publicly available transcriptomic datasets of humans (GSE137832 database; $n = 10$, healthy male; Fig. 7E) on *NR1D1*, *KMO* and *KYAT1* gene expressions in peripheral blood mononuclear cells (PBMCs). This study investigated the effects of two acute exercise protocols: an exhaustive session, performed at 80% of individual $V̇O_2$max until exhaustion, and a moderate session, conducted at 60% of $V̇O_2$max for a duration matched to the exhaustive protocol. Blood samples for gene expression measurements were analysed 1 h after exercise, and data from both exercise sessions were combined to enhance statistical analysis power.

### KYN enzymes' responses to knockout, overexpression and pharmacological activation of REV-ERB-α in C2C12 cells

The knockout of REV-ERB-α using the CRISPR/Cas9 approach increased KAT1 ($P = 0.001$) and KMO ($P < 0.0001$) protein levels (Fig. 8A), whereas over-

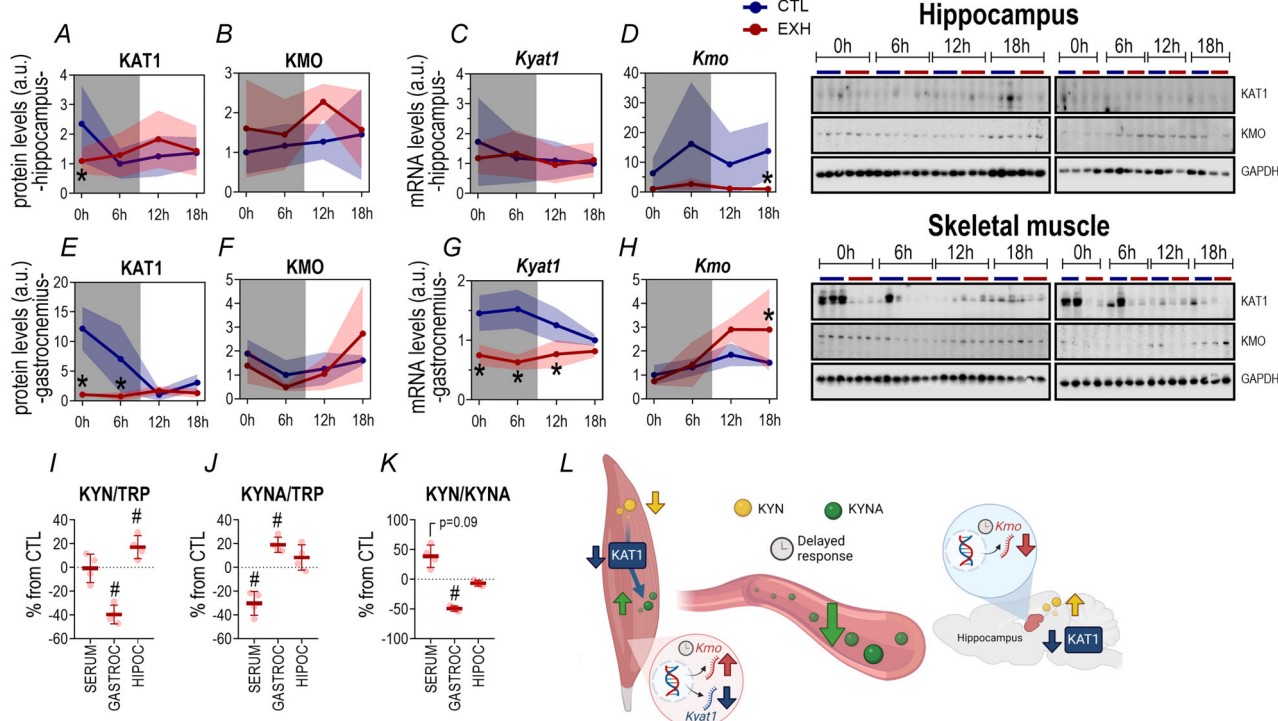

**Figure 5. Tissue-specific modulation of kynurenine pathway following acute exhaustive exercise**
Comparisons between groups of KAT1 (*A*), KMO (*B*) protein levels, *Kyat1* (*C*) and *Kmo* (*D*) mRNA levels in the hippocampus. Comparisons between groups of KAT1 (*E*), KMO (*F*) protein levels, *Kyat1* (*G*) and *Kmo* (*H*) mRNA levels in the gastrocnemius. Percentage alteration of kynurenine/tryptophan (*I*), kynurenic acid/tryptophan (*J*) and kynurenine/kynurenic acid ratios (*K*) immediately after the exhaustive exercise (EE) session compared to CTL in serum, gastrocnemius and hippocampus. Summary of the present data, highlighting the major responses of the KYN pathway to an acute exhaustive exercise stimulus (*L*). Data correspond to the mean ± SD of $n$ = 4–5 mice. *$P \leq 0.05$. CTL, control group; EXH, acute exhaustive exercise group. [Colour figure can be viewed at wileyonlinelibrary.com]

expression only reduced KMO protein levels ($P < 0.0001$; Fig. 8*B*). Treatment of the differentiated myotubes with SR9009 did not induce changes in the analysed proteins (Fig. 8*C*).

## REV-ERB-α agonist treatment effects in the KYN pathway

**Hippocampus: SR9009 reduced KMO protein levels and increased Kyat1 mRNA expression.** The SR9009 group

reduced KMO protein levels ($P = 0.038$; Fig. 9*A*) and increased *Kyat1* mRNA levels ($P = 0.043$; Fig. 9*B*) in the hippocampus compared to VEH. There were no alterations in the other evaluated parameters.

**Skeletal muscle: SR9009 reduced REV-ERB-α levels, accompanied by increased KAT1 and *Kyat1* levels.** The SR9009 group downregulated the REV-ERB-α ($P = 0.026$) and increased KAT1 ($P = 0.012$) protein levels in gastrocnemius compared to VEH (Fig. 9*C*). Also the same group

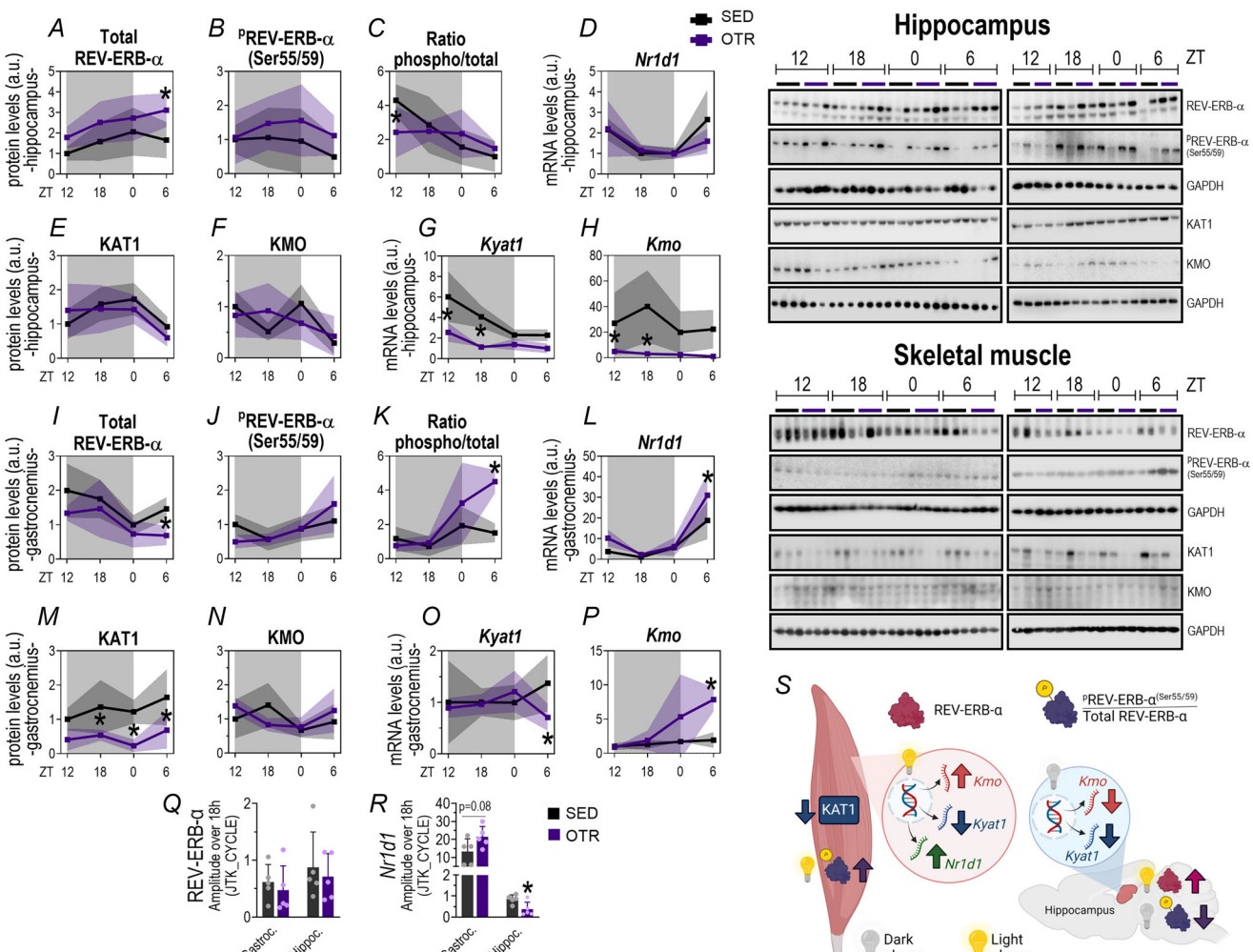

**Figure 6. REV-ERB-α and kynurenine pathway alterations in the hippocampus and skeletal muscle induced by chronic exhaustive exercise**

Comparisons between groups for total REV-ERB-α (*A*), ᵖREV-ERB-α ⁽ˢᵉʳ⁵⁵/⁵⁹⁾ (*B*), the ratio between phosphorylated/total REV-ERB-α (*C*) protein levels, *Nr1d1* mRNA levels (*D*), KAT1 (*E*) and KMO (*F*) protein levels, *Kyat1* (*G*) and *Kmo* (*H*) mRNA levels in the hippocampus. Comparisons between groups for total REV-ERB-α (*I*), ᵖREV-ERB-α ⁽ˢᵉʳ⁵⁵/⁵⁹⁾ (*J*), the ratio between phosphorylated/total REV-ERB-α (*K*) protein levels, *Nr1d1* mRNA levels (*L*), KAT1 (*M*) and KMO (*N*) protein levels, *Kyat1* (*O*) and *Kmo* (*P*) mRNA levels in gastrocnemius. Percentage alteration of kynurenine/tryptophan, kynurenic acid/tryptophan and kynurenine/kynurenic acid ratios (*Q*) at ZT12 of OTR mice compared to SED in serum. Amplitude analysis over 18 h through JTK_CYCLE for REV-ERB-α (*R*) and *Nr1d1* (*S*) in the gastrocnemius and hippocampus. Summary of the present data, highlighting the major responses of REV-ERB-α and KYN pathway to chronic exhaustive exercise training (*t*). Data correspond to the mean ± SD of $n = 4$–5 mice. *$P \leq 0.05$. SED, sedentary group; OTR, chronic exhaustive training group (overtraining), ZT, Zeitgeber time. [Colour figure can be viewed at wileyonlinelibrary.com]

upregulated *Kyat1* mRNA levels ($P = 0.001$; Fig. 9*D*) and increased KYN/KYNA ($P = 0.032$; Fig. 9*G*) ratios.

**Serum: SR9009 treatment increased KYN levels.** The SR9009 group increased the KYN/TRP ratio in serum when compared to VEH ($P = 0.007$; Fig. 9*E*). Also this group showed a tendency ($P = 0.06$) to a higher ratio between KYN and KYNA, suggesting a potential neurotoxic imbalance (Fig. 9*G*).

## Discussion

The main findings of this study were as follows: (a) acute EE abolished the diurnal rhythm of REV-ERB-α in skeletal muscle, downregulating its nuclear and cytosolic levels; (b) acute EE increased the phosphorylated/total ratio of REV-ERB-α within the first 6 h in both skeletal muscle and hippocampus; (c) acute EE reduced KAT1 content peripherally and centrally, while upregulating *Kmo* mRNA; (d) acute EE reduced KYN bioavailability (KYN/TRP ratio) in skeletal muscle but increased it in the hippocampus; (e) REV-ERB-α negatively correlated with KMO protein levels, whereas *Nr1d1* positively correlated with *Kmo* mRNA in the gastrocnemius in response to acute EE; (f) the OT protocol modulated REV-ERB-α/*Nr1d1* expression patterns in skeletal muscle and/or hippocampus; (g) OT decreased KAT1/*Kyat1* expression in both tissues, upregulated *Kmo* in skeletal muscle and downregulated it in the hippocampus; (h) in C2C12 muscle cells REV-ERB-α KO increased KAT1 and KMO protein levels, whereas overexpression of REV-ERB-α reduced KMO protein levels; (i) treatment with an REV-ERB-α agonist increased KAT1 levels in skeletal muscle and reduced KMO protein in the hippocampus; and (j) SR9009 treatment increased KYN bioavailability in serum but decreased it in skeletal muscle and hippocampus.

### REV-ERB-α and acute exhaustive exercise

Acute EE activates biochemical pathways associated with proinflammatory responses and immune activation in trained athletes, leading to TRP catabolism and increased KYN pathway activation in the blood (Strasser et al.,

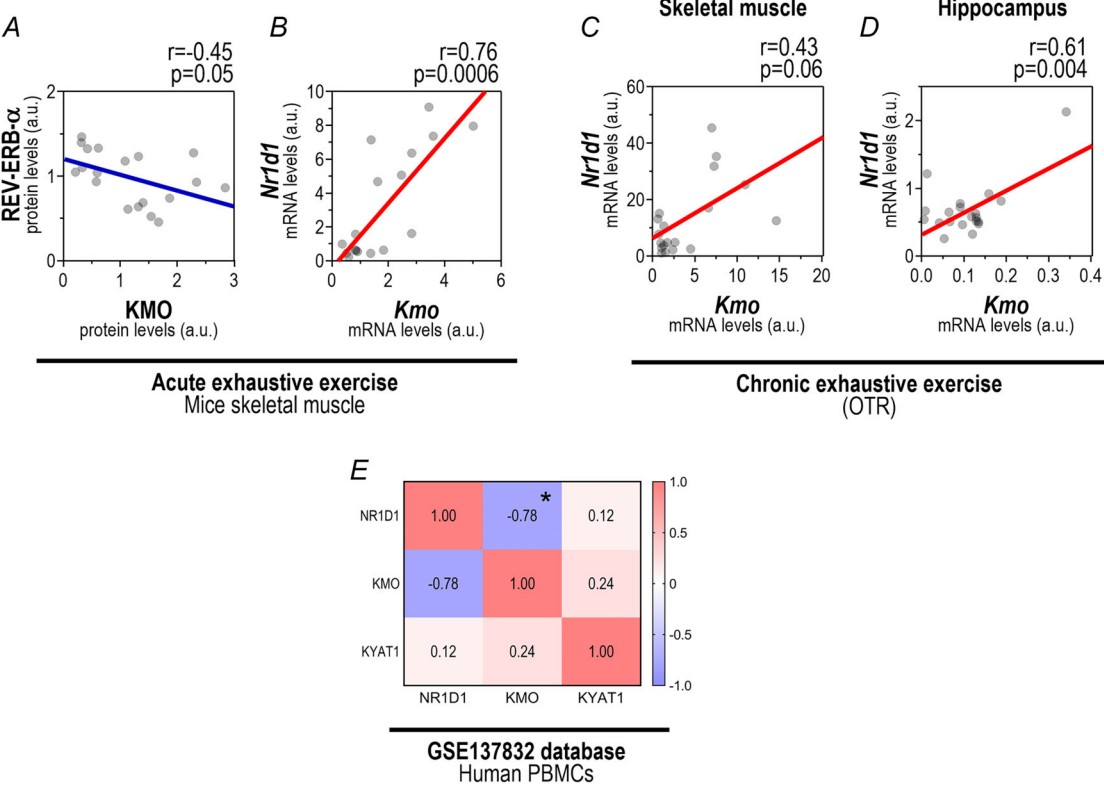

**Figure 7. Correlations between REV-ERB-α and kynurenine pathway components in mouse and human tissues**
Correlation between total REV-ERB-α and KMO protein levels (*A*) and between *Nr1d1* and *Kmo* mRNA levels (*B*) in the gastrocnemius muscle of the EXH group. Correlations between *Nr1d1* and *Kmo* mRNA levels in the gastrocnemius (*C*) and hippocampus (*D*) of the OTR group. (*E*) Heatmap showing *NR1D1*, *KMO* and *KYAT1* correlations from publicly available human transcriptomic datasets (GSE137832; *n* = 10). [Colour figure can be viewed at wileyonlinelibrary.com]

2016). The authors suggested that these responses might contribute to fatigue and mood disturbances. In contrast da Rocha et al. (2019) reported that chronic EE led to inflammation in peripheral tissues, potentially impairing physical performance and causing prolonged fatigue. Recently the same group identified REV-ERB-$\alpha$ as a key regulator of exercise adaptations, noting its high responsiveness to different exercise models (da Rocha et al., 2022). Global ablation of this protein reduces skeletal muscle oxidative capacity and causes exercise intolerance (Woldt et al., 2013).

In this study acute EE reduced REV-ERB-$\alpha$ levels in the gastrocnemius and abolished its diurnal oscillation, reducing the expression amplitude for 18 h postexercise. Surprisingly both nuclear and cytoplasmic REV-ERB-$\alpha$ levels decreased in skeletal muscle. Physical exercise can activate the ubiquitin-proteasome pathway (UPP) in skeletal muscle, with eccentric contraction-based exercises showing the most significant effects (Reid, 2005). The UPP responds to exercise in two phases: an initial activation within seconds to minutes of exercise onset and a delayed increase peaking between 6 and 24 h. This delayed response supports muscle remodelling by regulating protein degradation during adaptation (Reid, 2005). Under exhaustive exercise this response intensifies, leading to reductions in nuclear and cytoplasmic content due to the predominance of eccentric contractions and high energy demands.

On the contrary Small et al. (2020) demonstrated that exercise directly regulates skeletal muscle clock gene expression via muscle contraction using a calcium-dependent pathway. Specifically analysis of human vastus lateralis biopsies collected 1 h after an intense exercise session (15 min at 80% $VO_2$max) revealed decreased expression of *NR1D1* and increased expression of *PER1* and *PER2*. Additional studies are necessary to elucidate the precise mechanisms by which exercise modulates REV-ERB-$\alpha$/*Nr1d1* expression.

Phosphorylated REV-ERB-$\alpha$ showed an inverse pattern and correlation to total REV-ERB-$\alpha$ during the diurnal rhythm in skeletal muscle under basal conditions (Fig. 4*A*). This newly observed dynamic may shed light on the regulatory mechanisms of this protein, highlighting a compelling link between circadian oscillation and the post-translational modification of REV-ERB-$\alpha$. As noted phosphorylation at serine 55/59 by glycogen synthase kinase 3$\beta$ (GSK3$\beta$) enhances its stability (Yin, 2004). REV-ERB-$\alpha$ belongs to a group of proteins (Clock, Bmal1, ROR, Per and Cry) that self-regulate through transcriptional negative feedback loops, creating 24 h oscillations in their total concentrations (Solt et al., 2011). Post-translational phosphorylation of REV-ERB-$\alpha$ in skeletal muscle appears to act as a sensor for its bioavailability. The inverse relationship between total and phosphorylated protein levels during the diurnal oscillation suggests that when total REV-ERB-$\alpha$ decreases, phosphorylation

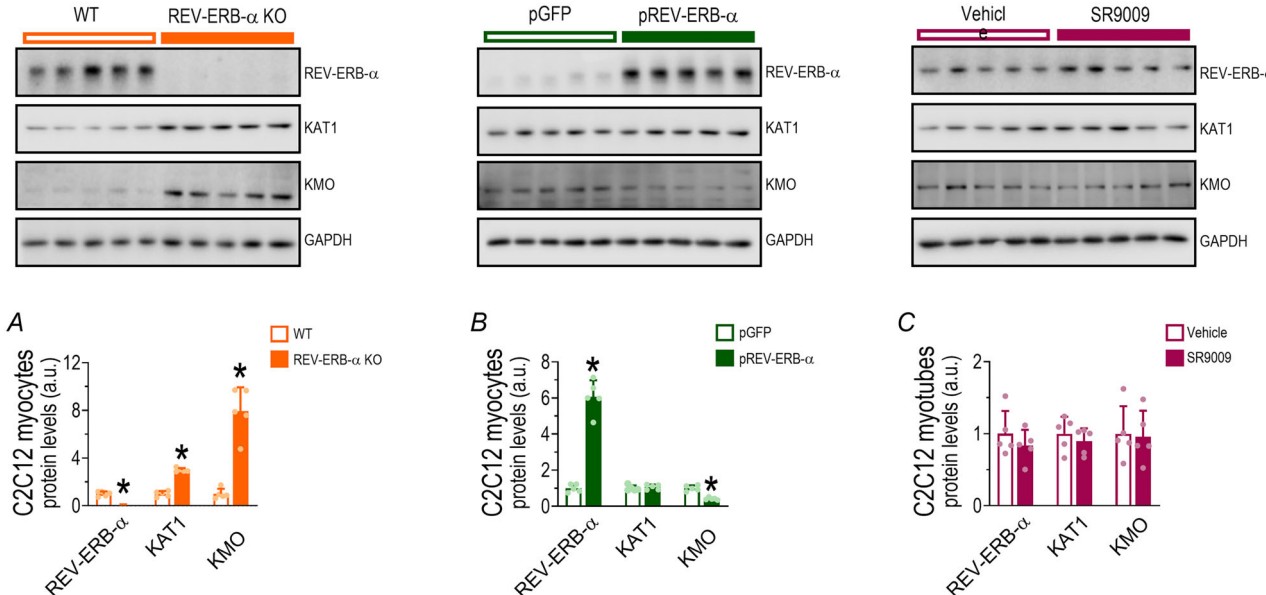

**Figure 8. Effects of REV-ERB-$\alpha$ knockout, overexpression, and pharmacological activation on kynurenine pathway in C2C12 cells**
Protein levels of REV-ERB-$\alpha$, KAT1 and KMO in C2C12 REV-ERB-$\alpha$ knockout myoblasts (*A*), C2C12 REV-ERB-$\alpha$ overexpression myoblasts (*B*) and C2C12 differentiated myotubes treated with SR9009 (*C*). *$P < 0.05$ *vs.* control group of the experiment. Data correspond to the mean ± SD of *n* = 5 per group. [Colour figure can be viewed at wileyonlinelibrary.com]

increases to maintain viable levels in the cell. Thus the phosphorylated to total REV-ERB-α ratio may indicate its availability or activity. However further studies are needed to clarify this relationship and validate this theory.

Acute EE led to a fivefold increase in REV-ERB-α phosphorylation in the hippocampus immediately after the session, without a corresponding reduction in total REV-ERB-α, as observed in skeletal muscle. This discrepancy may be attributed to differences in stress stimuli. Skeletal muscle is exposed to various metabolites and environmental conditions (Morton et al., 2009), along with mechanical stress from eccentric

contractions that can induce tissue damage, triggering local responses such as protein degradation through the UPP (Murton et al., 2008) and activation of a calcium-dependent signalling pathway mediated by muscle contraction (Small et al., 2020). In contrast the brain is protected by the blood–brain barrier (BBB), which stabilizes the CNS environment by regulating the entry of potentially harmful substances and metabolites (Małkiewicz et al., 2019). Interestingly the ratio of phosphorylated to total REV-ERB-α in both tissues shows a similar response in magnitude and timing. However this occurs through different mechanisms, with

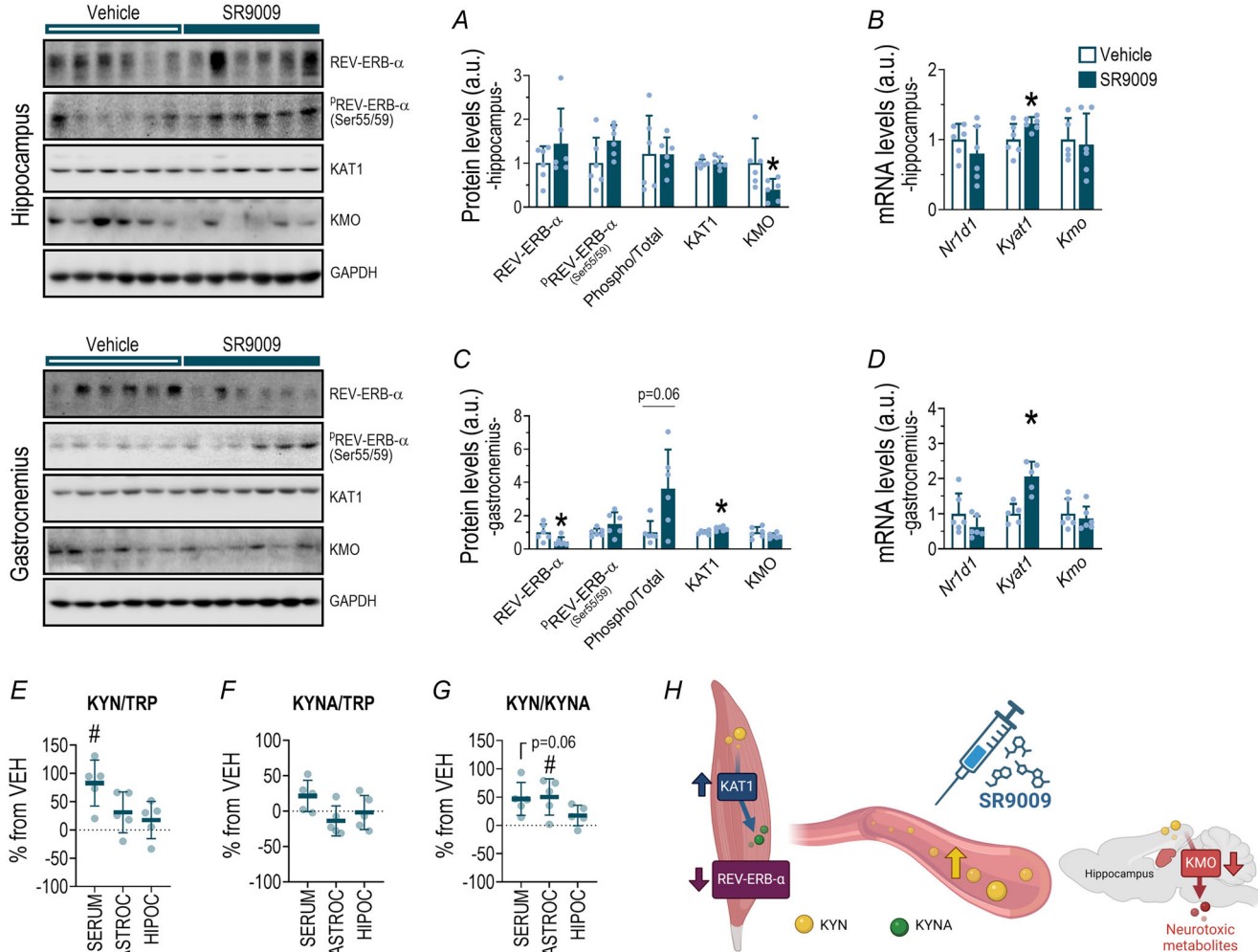

**Figure 9. Effects of REV-ERB-α agonist SR9009 on kynurenine pathway regulation in hippocampus, skeletal muscle, and serum**
Protein levels of REV-ERB-α, ᵖREV-ERB-α$^{(Ser55/59)}$, ᵖREV-ERB-α$^{(Ser55/59)}$/REV-ERB-α ratio, KAT1 and KMO in hippocampus (*A*). mRNA levels of *Nr1d1*, *Kyat1* and *Kmo* in the hippocampus (*B*). Protein levels of REV-ERB-α, ᵖREV-ERB-α$^{(Ser55/59)}$, ᵖREV-ERB-α$^{(Ser55/59)}$ / REV-ERB-α ratio, KAT1 and KMO in gastrocnemius (*C*). mRNA levels of *Nr1d1*, *Kyat1* and *Kmo* in the gastrocnemius (*D*). Percentage alteration of kynurenine/typtophan (*E*), kynurenic acid/tryptophan (*F*) and kynurenine/kynurenic acid ratios (*G*) after SR9009 treatment compared to VEH in serum, gastrocnemius and hippocampus. Summary of the present data, highlighting the major responses of REV-ERB-α and the KYN pathway to SR9009 treatment (*H*). Data correspond to the mean ± SD of *n* = 4–6 mice. *$P \leq 0.05$. VEH, vehicle treated group; SR9009, REV-ERB-α agonist treated group. [Colour figure can be viewed at wileyonlinelibrary.com]

decreased total REV-ERB-$\alpha$ in muscle and increased phosphorylation in the hippocampus.

## Kynurenine pathway and acute exhaustive exercise

The KYN pathway produces intermediate metabolites with neurotoxic effects, such as neuronal death and neuroinflammation (Myint & Kim, 2014), which can be triggered by stress and inflammation (Gibney et al., 2013). EE is highly stressful for metabolism and is linked to an inflammatory profile (da Rocha et al., 2019). In this study acute EE increased KYN bioavailability in the hippocampus relative to TRP (the precursor of serotonin and KYN) and decreased KYN levels in skeletal muscle immediately postexercise. Notably EE suppressed KAT1 protein in both skeletal muscle and the hippocampus. KATs convert KYN into KYNA, a neuroprotective metabolite that cannot cross the BBB (Fukui et al., 1991) and exerts anti-inflammatory effects on immune cells (Cervenka et al., 2017). In the hippocampus this increase in KYN bioavailability may explain the suppression of *Kmo* expression, which encodes KMO, responsible for converting KYN into 3-HK, a neurotoxic metabolite, suggesting a protective strategy due to increased kynurenine levels.

Regular physical exercise is closely linked to KYN metabolism in skeletal muscle and neural protection. For instance Joisten et al. (2020) demonstrated that an acute exercise bout induces alterations in the KYN pathway in human serum and PBMCs samples, highlighting differential responses between endurance and resistance exercise models. Agudelo and colleagues (Agudelo et al., 2014) showed that exercise increases KAT expression in skeletal muscle by activating peroxisome proliferator-activated receptor gamma coactivator 1 alpha (PGC-1$\alpha$). Moreover PGC-1$\alpha$ overexpression in muscle elevated KAT levels and prevented the development of KYN-induced neurological disorders. Therefore the suppression of KAT1 in skeletal muscle caused by acute EE, combined with increased *Kmo* gene expression, indicates a shift towards producing neurotoxic metabolites via the KYN pathway.

Our findings demonstrate that acute EE significantly impacts skeletal muscle and hippocampal REV-ERB-$\alpha$, as well as components of the KYN pathway. In skeletal muscle we observed a negative correlation between REV-ERB-$\alpha$ and protein levels of KMO, a key enzyme in the KYN pathway. Conversely *Nr1d1* mRNA exhibited a positive correlation with *Kmo* transcription. Although this might initially appear contradictory, it reflects the regulatory dynamics between the REV-ERB-$\alpha$ protein and *Nr1d1* mRNA expression. As expected REV-ERB-$\alpha$ protein exhibits an inverted rhythmic pattern relative to *Nr1d1* transcription in skeletal muscle, such that peak

protein abundance coincides with minimal *Nr1d1* mRNA expression, and vice versa, as shown in baseline skeletal muscle samples (Fig. 4*A* and *B*). This inverse relationship, characteristic of transcriptional–translational feedback loops, helps to clarify the apparent discrepancy between mRNA and protein-level correlations (Robles et al., 2014). Collectively these data suggest that REV-ERB-$\alpha$ may exert transcriptional repression on the KMO enzyme, consistent with its established role in circadian rhythm regulation.

Interestingly bioinformatic analysis of the GSE137832 database supports our findings by revealing an inverse relationship between these two molecules. Although the database indicated an inverse correlation between *Nr1d1* and *Kmo* mRNA, it is important to emphasize that those data were derived from PBMCs, whereas our study analysed skeletal muscle tissue. This distinction is critical given that REV-ERB-$\alpha$/*Nr1d1* expression patterns are known to exhibit tissue-specific circadian rhythms (Everett & Lazar, 2014). Furthermore the GSE137892 database relied on a single postexercise time point, whereas our study adopted a time course design, offering a broader temporal perspective. Additionally mice have an inverted wake–rest cycle compared to humans, which influences *Nr1d1* circadian rhythm (Gutierrez-Monreal et al., 2020). Despite these methodological and physiological differences, the observed correlations remained consistent across both datasets.

Parrott et al. (2016) identified KMO as a key enzyme in KYN metabolism within the hippocampus. Using wild-type animals treated with LPS to induce inflammation and depressive behaviour, they found that the KYN pathway was particularly activated in the hippocampus. Interestingly KMO KO animals were protected from LPS-induced depressive behaviour. The KYN pathway is gaining attention as a potential mechanism in the aetiology of neurological disorders, including Alzheimer's, Parkinson's and Huntington's diseases (Braidy & Grant, 2017).

Collectively the increased bioavailability of KYN in the hippocampus, the reduced KAT1 levels in both peripheral and central tissues and the upregulation of *Kmo* in skeletal muscle suggest a shift towards the neurotoxic branch of the KYN pathway. The downregulation of KAT1 impairs the conversion of KYN to KYNA, increasing its availability for conversion into 3-HK and QUIN. Both metabolites can cross the BBB and contribute to oxidative stress, lipid peroxidation and apoptosis (Okuda et al., 1998).

## Overtraining, REV-ERB-$\alpha$ and KYN pathway

Knowing chronic EE can lead to extreme fatigue (Meeusen et al., 2013), we assessed whether the OT protocol modulates REV-ERB-$\alpha$ at the transcriptional and protein

levels throughout the diurnal oscillation. It is important to note that the analyses were performed 48 h after the last physical test, a timing that likely influenced the observed results. The acute response (as assessed in Experiment 1) reflects the immediate effects of the exercise stimulus, whereas the samples analysed in the chronic protocol (Experiment 2) capture the adaptation phase. Skeletal muscle and hippocampus showed distinct responses in this case. In skeletal muscle the lower content of REV-ERB-α suggests maladaptation to the OT protocol. Woldt et al. (2013) found that moderate training for 4 weeks increased physical performance along with REV-ERB-α content in the gastrocnemius, whereas genetic deletion of REV-ERB-α had the opposite effect. This dysregulation of REV-ERB-α's cycle may indicate disturbances in physiological rhythms, potentially linked to metabolic diseases (Gabriel & Zierath, 2019) and reduced physical performance.

Sgro et al. (2023) showed that repetitive mild traumatic brain injury alters the expression of central clock genes in several brain regions, including the hippocampus, potentially disrupting cellular processes and increasing the risk of psychiatric disorders. Similarly Bering et al. (2023) demonstrated that rhythmic corticosterone infusion in mice reduced *Nr1d1* expression in the *dentate gyrus* and CA1 region of the hippocampus. It is worth noting that physical exercise can double corticosterone levels in Wistar rats (Ebal et al., 2007), highlighting this interrelationship between physical exercise and corticosterone, as well as its possible influence on the CNS. These findings suggest chronic EE may reduce *Nr1d1* expression through elevated corticosterone levels. However further studies are needed to confirm this theory.

Although the OT protocol did not alter serum levels of TRP, KYN or KYNA, it induced changes in KYN pathway enzymes similar to those observed after a single exhaustive exercise session. In the gastrocnemius muscle the suppression of KAT1 at both transcriptional and protein levels indicates a reduced capacity for neural protection under conditions like stress and inflammation (Gibney et al., 2013). This adaptation contrasts with the benefits of regular moderate-intensity exercise (Agudelo et al., 2014). The increased *Kmo* expression further suggests maladaptation, making the skeletal muscle more inclined to metabolize KYN through the neurotoxic pathway, potentially releasing intermediate KYNs into the bloodstream that can cross the BBB.

It has already been established that excessive training can also induce negative adaptations in the liver, an organ that plays a central role in systemic metabolism (da Rocha et al., 2019). In mice subjected to excessive training the liver displays increased inflammatory markers and fat accumulation (da Rocha et al., 2017). Given its significant influence on metabolism and sensitivity to stress-related factors such as proinflammatory cytokines, the liver may

also play a key role in regulating metabolites derived from the KYN pathway. Indeed under basal conditions the liver metabolizes approximately 90% of tryptophan through the KYN pathway and expresses all the enzymes necessary for this metabolic route (Badawy, 2017). Therefore future studies should investigate the effects of excessive training on the regulation of KYN pathway enzymes in the liver to better understand its contribution to peripheral metabolite regulation and their bioavailability to the CNS.

In the hippocampus the KYN pathway showed suppression of genes encoding enzymes involved in its metabolism, possibly as a neuroprotective strategy. Dysregulation of this pathway, linked to inflammation, can lead to CNS maladaptations, such as depression (Dantzer et al., 2011). Recently the KYN pathway has been identified as a promising target for treating major depressive disorders (Brown et al., 2021). Symptoms of OT syndrome in athletes, such as apathy, fatigue and sleep disturbances, often resemble those of depression (Dantzer et al., 2011). Notably depression incidence is higher among athletes compared to the general population. For example Golding and coworkers (Golding et al., 2020) reported a prevalence of depressive symptoms reaching up to 34% in high-performance athletes. Although depression has a multifactorial aetiology, further investigation into the KYN pathway is warranted to understand better the effects of chronic exhaustive exercise on the CNS.

Indeed the activation of the KYN pathway and the subsequent production of neurotoxic metabolites can contribute to the development of neurological disorders through multiple mechanisms. One such metabolite, 3-HK, is a potent generator of free radicals and has been found at elevated levels in several neurodegenerative diseases, including Huntington's disease and Alzheimer's disease (Guidetti et al., 2006; Sharma et al., 2022). Both *in vitro* and *in vivo* studies have demonstrated that 3-HK can induce neuronal cell death via apoptotic pathways (Eastman & Guilarte, 1989). Furthermore 3-HK promotes oxidative stress, leading to lipid peroxidation and consequent cell damage (Schwarcz et al., 2012). Another key neurotoxic metabolite, QUIN, functions as an agonist of *N*-methyl-D-aspartate (NMDA) receptors and possesses strong excitotoxic properties by increasing glutamate levels and promoting neurotoxicity, particularly within the hippocampus (Pittaluga et al., 2001). In addition to its excitotoxic effects QUIN can also induce lipid peroxidation and has been shown to disrupt the integrity of the BBB (Behan & Stone, 2002). Summarizing the KYN pathway requires highly sensitive regulatory control, as imbalances in its activity can have deleterious consequences for the CNS.

Overall acute and chronic exhaustive exercise protocols resulted in the downregulation of KAT1 and/or *Kyat1* in peripheral and central tissues, accompanied by

upregulation of *Kmo* in skeletal muscle. This expression pattern suggests a maladaptive response triggered by the initial exhaustive exercise bout and sustained throughout the 8-week OT protocol. Skeletal muscle is directly impacted by downhill running, which can induce tissue damage. Without sufficient recovery periods, muscle injury is perpetuated, continually reinforcing the detrimental stimulus. Conversely the CNS experiences indirect effects mediated by factors released peripherally during exhaustive exercise. Initially the acute response may reflect a protective mechanism to stabilize CNS homeostasis. However repeated exposure during chronic exhaustive exercise likely leads to metabolic dysregulation within the CNS. Indeed Chung and coworkers (Chung et al., 2021) demonstrated that an 8-week overtraining protocol impaired metabolic function and induced

anxiolytic-like behaviour in mice. Interestingly Rosa et al. (2007) also reported cognitive impairment after only 10 days of exhaustive training in mice.

## Interplay between REV-ERB-α and KYN pathway

To test whether REV-ERB-α modulates KYN pathway enzymes we genetically manipulated C2C12 muscle cells. Surprisingly in the absence of REV-ERB-α, both KAT1 and KMO showed marked upregulation, whereas over-expression of REV-ERB-α downregulated only KMO. These results suggest that REV-ERB-α acts as a transcriptional repressor of these enzymes, particularly KMO. This finding enhances our understanding of the interplay between these pathways and their outcomes. REV-ERB-α

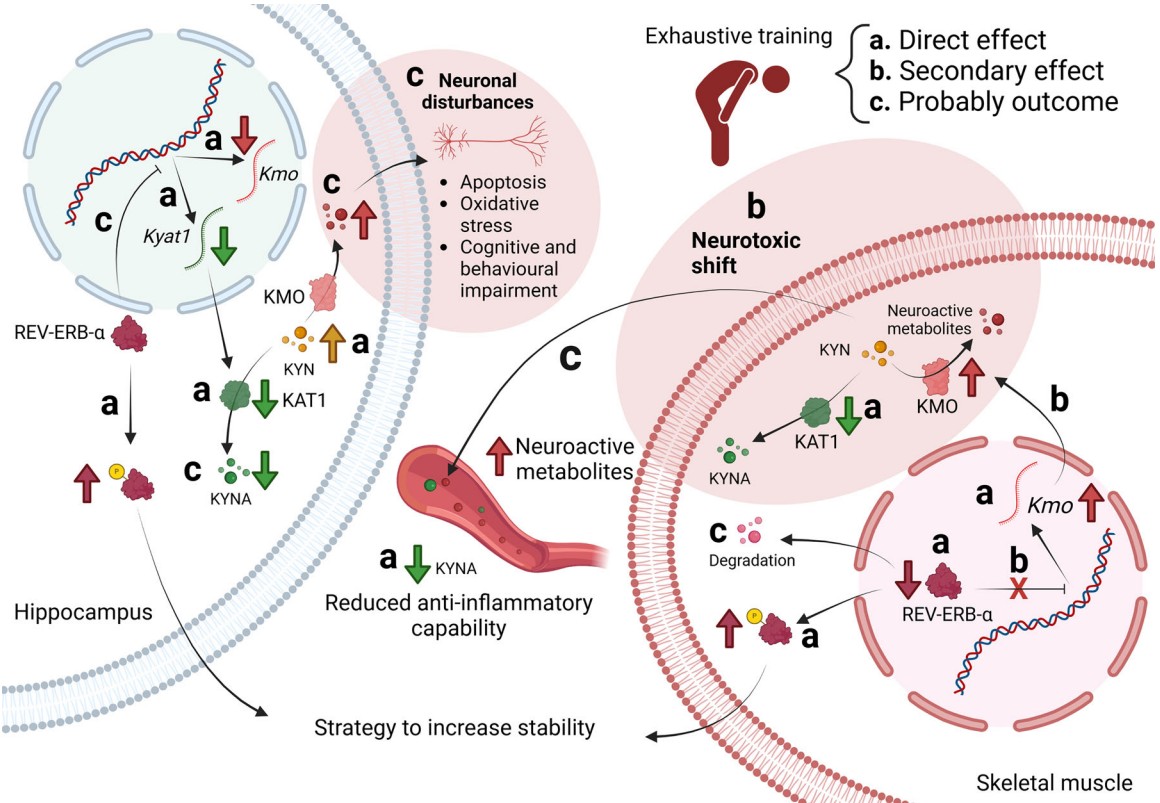

**Figure 10. Proposed mechanism of the maladaptation induced by exhaustive exercise**
Exhaustive exercise causes a reduction in REV-ERB-α in skeletal muscle, leading to increased phosphorylation levels to enhance its stability in response to low protein levels. The reduction in total REV-ERB-α decreases its ability to repress KMO transcription, expanding the availability of its mRNA to form active proteins and metabolize kynurenine (KYN) into neuroactive metabolites that can be released into the bloodstream. Additionally exhaustive exercise reduces KAT1 levels, decreasing the capability to metabolize KYN into kynurenic acid (KYNA) in the bloodstream. This shift highlights a neurotoxic tendency, with lower levels of KYNA and probably higher levels of neuroactive metabolites in serum that can cross the blood–brain barrier. In the hippocampus exhaustive exercise increased the phosphorylation of REV-ERB-α and reduced the transcription of KAT1 and KMO, possibly aiming at neuroprotection of the tissue due to higher levels of KYN. However only KAT1 protein levels were reduced, which impairs the metabolization of KYN into KYNA. Therefore the accumulated levels of KYN in the hippocampus probably increase its metabolization through KMO, producing neuroactive metabolites that lead to neuronal disturbances, ultimately provoking cognitive and behavioural disorders, such as depression. Created in BioRender. Da Rocha, A. (2025). [Colour figure can be viewed at wileyonlinelibrary.com]

is a potential target in several diseases, regulating metabolism under circadian rhythm. For example it downregulates toll-like receptor 4 (TLR4) and reduces IL-6 production, thereby controlling inflammation in the immune system (Scheiermann et al., 2013). REV-ERB-α plays a complex role in skeletal muscle, producing positive effects when activated or inhibited, depending on the context (Uriz-Huarte et al., 2020). These findings underscore the critical role of REV-ERB-α as a transcriptional repressor in various cellular processes.

Although myotubes did not respond to treatment with the REV-ERB-α agonist SR9009, administration in C57BL/6 mice revealed significant metabolic effects *in vivo*. SR9009 increased the serum KYN/TRP and KYN/KYNA ratios in skeletal muscle. Additionally it elevated KAT1 content in skeletal muscle and reduced KMO protein levels, specifically in the hippocampus. These findings suggest that pharmacological activation of REV-ERB-α may enhance skeletal muscle's ability to metabolize KYN to KYNA, preventing its accumulation in the CNS and highlighting a neuroprotective role. This effect has been observed only with regular physical exercise (Schlittler et al., 2016). For the first time we demonstrated that SR9009 reduces KMO, a key KYN pathway enzyme in the hippocampus that produces neurotoxic metabolites. This supports the REV-ERB-α and KMO interaction hypothesis, suggesting that it is a potential therapeutic target for regulating this pathway in the hippocampus.

### Limitations and future directions

Although this study provides valuable insights into the molecular responses to EE, several limitations must be acknowledged. First this research exclusively assessed responses to EE; thus future studies should evaluate the effects of moderate and appropriate exercise intensities, as well as other modalities (e.g. endurance, strength, combined and interval training), to better elucidate the relationship between *Nr1d1* and the KYN pathway. Additionally analysing muscle groups with distinct fibre-type compositions (fast- *vs.* slow-twitch) may offer deeper insights into fibre-type specificity and its molecular adaptations to exercise.

Furthermore investigating other KYN metabolites (e.g. 3-HK, QUIN and PIC) would provide a more comprehensive understanding of EE-induced metabolic changes and their implications for different CNS regions. Finally integrating analyses of other circadian clock components with behavioural and cognitive assessments could further clarify the potential neurological impairments associated with EE. Collectively these limitations represent important avenues for future research to refine and expand upon our findings.

### Conclusions

In summary acute EE and OT led to robust modulations of REV-ERB-α, particularly evident in skeletal muscle, suggesting a potential association with fatigue and performance decline. Furthermore the exhaustive exercise models (acute and chronic) increased *Kmo* expression, the initial enzyme in the neurotoxic branch of the KYN pathway, concurrently with a decrease in KAT1, which typically exerts neuroprotective effects within skeletal muscle. This observed shift implies a maladaptive response in the skeletal muscle that may result in the efflux of neuroactive KYNs into the bloodstream, potentially reaching the CNS, given that certain metabolites can traverse the BBB. Notably this study establishes, for the first time, a correlation between REV-ERB-α and KMO, an enzyme within the KYN pathway. This interplay suggests a potential therapeutic target to minimize the harmful effects of exhaustive training. Figure 10 provides a graphical representation of the principal findings derived from the present investigation.

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

## Additional information

### Data availability statement

Data supporting the results of this study can be obtained from the corresponding author upon reasonable request.

### Competing interests

The authors declare no competing interests.

### Author contributions

All authors approved the final version of the manuscript and agreed to be accountable for all aspects of the work to ensure that questions related to the accuracy or integrity of any part of the work are appropriately investigated and resolved. All persons designated as authors qualify for authorship, and all those who qualify are listed. Conceptualization: A.L.R. and A.S.R.S.; methodology: A.L.R., A.P.P., V.R.M., B.B.M. and L.E.C.M.S.; validation: A.L.R., A.P.P., B.B.M., I.V.S.N., V.R.M., L.E.C.M.S., J.R.P., D.E.C., E.R.R., F.M.S., L.P.M., E.C.F., D.A.R. and A.S.R.S.; formal analysis: A.L.R., A.P.P., I.V.S.N., V.R.M., B.B.M. and L.E.C.M.S.; investigation: A.L.R., A.P.P., V.R.M. and B.B.M.; resources: J.R.P., D.E.C., E.R.R., F.M.S., L.P.M., E.C.F., D.A.R. and A.S.R.S.; data curation: A.L.R., A.P.P. and I.V.S.N; writing – original draft preparation: A.L.R. and A.S.R.S.; visualization: A.L.R., A.P.P., I.V.S.N., V.R.M., B.B.M., L.E.C.M.S., J.R.P., D.E.C., E.R.R., F.M.S., L.P.M., E.C.F., D.A.R. and A.S.R.S.; supervision: A.S.R.D.S.; project administration: A.L.R.

### Funding

The present work received financial support from the São Paulo Research Foundation (FAPESP; process numbers 17/12765-2, 19/11820-5, 21/08693-1 and 22/06807-2), the National Council for Scientific and Technological Development (CNPq; process numbers 301279/2019-5 and 308999/2022-3) and the Coordination for the Improvement of Higher Education Personnel (CAPES; finance code 001).

### Acknowledgements

We thank Dr. Jorge L. Ruas for his valuable intellectual contributions and insightful discussions throughout the development of this work.

### Keywords

exhaustive exercise, fatigue, kynurenine, neuromodulation, SR9009

## Supporting information

Additional supporting information can be found online in the Supporting Information section at the end of the HTML view of the article. Supporting information files available:

**Peer Review History**

