## [Peer Review History · The Journal of Physiology]

EXHAUSTIVE EXERCISE ABOLISHES REV-ERB- α CIRCADIAN RHYTHM AND SHIFTS THE KYNURENINE PATHWAY TO A NEUROTOXIC PROFILE IN MICE

Alisson Luiz da Rocha, Ana Paula Pinto, Ivo Vieira de Sousa Neto, Vitor Rosetto Munoz, Bruno Brieda Marafon, Lilian Eslaine Costa Mendes da Silva, José Rodrigo Pauli, Dennys Esper Cintra, Eduardo R Ropelle, Fernando Simabuco, Leandro P de Moura, Ellen Cristini de Freitas, Donato Americo Rivas, and Adelino Sanchez Ramos da Silva Silva
DOI: 10.1113/JP288290

Corresponding author(s): Alisson da Rocha (alisson.rocha@alumni.usp.br)

Review Timeline:

Submission Date:	02-Dec-2024
Editorial Decision:	27-Jan-2025
Revision Received:	10-Apr-2025
Editorial Decision:	07-May-2025
Revision Received:	20-May-2025
Accepted:	04-Jun-2025

Senior Editor: Karyn Hamilton

Reviewing Editor: Mark Viggars

Transaction Report:

Dear Dr da Rocha,

Re: JP-RP-2024-288290 "EXHAUSTIVE EXERCISE ABOLISHES REV-ERB- α CIRCADIAN RHYTHM AND SHIFTS THE KYNURENINE PATHWAY TO A NEUROTOXIC PROFILE IN MICE" by Alisson Luiz da Rocha, Ana Paula Pinto, Ivo Vieira de Sousa Neto, Vitor Rosetto Munoz, Bruno Brieda Marafon, Lilian Esleine Costa Mendes da Silva, José Rodrigo Pauli, Dennys Esper Cintra, Eduardo R Ropelle, Fernando Simabuco, Leandro P de Moura, Ellen Cristini de Freitas, Donato Americo Rivas, and Adelino Sanchez Ramos da Silva Silva

Thank you for submitting your manuscript to The Journal of Physiology. It has been assessed by a Reviewing Editor and by 2 expert referees and we are pleased to tell you that it is potentially acceptable for publication following satisfactory major revision.

REVISION CHECKLIST:

Upload a full Response to Referees file. To create your 'Response to Referees': copy all the reports, including any comments from the Senior and Reviewing Editors, into a Microsoft Word, or similar, file and respond to each point, using

font or background colour to distinguish comments and responses and upload as the required file type.

We look forward to receiving your revised submission.

Yours sincerely,

Karyn Hamilton
Senior Editor
The Journal of Physiology

REQUIRED ITEMS

- Include a Key Points list in the article itself, before the Abstract.
- Author photo and profile. First or joint first authors are asked to provide a short biography (no more than 100 words for one author or 150 words in total for joint first authors) and a portrait photograph. These should be uploaded and clearly labelled together in a Word document with the revised version of the manuscript. See Information for Authors for further details.
- The reference list must be in alphabetical order, rather than numbered, to comply with our Journal format.
- Your manuscript must include a complete Additional Information section, including competing interests; funding; author contributions and acknowledgements.
- The Journal of Physiology funds authors of provisionally accepted papers to use the premium BioRender site to create high resolution schematic figures. Follow this link and enter your details and the manuscript number to create and download figures. Upload these as the figure files for your revised submission. If you choose not to take up this offer, we require figures to be of similar quality and resolution. If you are opting out of this service to authors, state this in the Comments section on the Detailed Information page of the submission form. The link provided should only be used for the purposes of this submission. Authors will be charged for figures created on this premium BioRender account if they are not related to this manuscript submission.
- Please upload separate high-quality figure files via the submission form.

- You must upload original, uncropped western blot/gel images (including controls) if they are not included in the manuscript. This is to confirm that no inappropriate, unethical or misleading image manipulation has occurred. These should be uploaded as 'Supporting information for review process only'. Please label/highlight the original gels so that we can clearly see which sections/lanes have been used in the manuscript figures. For more information, see: <https://physoc.onlinelibrary.wiley.com/hub/journal-policies#imagmanip>.

- Your paper contains Supporting Information of a type that we no longer publish, including supplementary tables and figures. Any information essential to an understanding of the paper must be included as part of the main manuscript and figures. The only Supporting Information that we publish are video and audio, 3D structures, program codes and large data files. Your revised paper will be returned to you if it does not adhere to our Supporting Information Guidelines.

- A Data Availability Statement is required for all papers reporting original data. This must be in the Additional Information section of the manuscript itself. It must have the paragraph heading 'Data Availability Statement'. All data supporting the results in the paper must be either: in the paper itself; uploaded as Supporting Information for Online Publication; or archived in an appropriate public repository. The statement needs to describe the availability or the absence of shared data. Authors must include in their statement: a link to the repository they have used, or a statement that it is available as Supporting Information; reference the data in the appropriate section(s) of their manuscript; and cite the data they have shared in the References section. Whenever possible, the scripts and other artefacts used to generate the analyses presented in the paper should also be publicly archived. If sharing data compromises ethical standards or legal requirements then authors are not expected to share it, but must note this in their statement. For more information, see our Statistics Policy.

- Please include an Abstract Figure file, as well as the Figure Legend text within the main article file. The Abstract Figure is a piece of artwork designed to give readers an immediate understanding of the research and should summarise the main conclusions. If possible, the image should be easily 'readable' from left to right or top to bottom. It should show the physiological relevance of the manuscript so readers can assess the importance and content of its findings. Abstract Figures should not merely recapitulate other figures in the manuscript. Please try to keep the diagram as simple as possible and without superfluous information that may distract from the main conclusion(s). Abstract Figures must be provided by authors no later than the revised manuscript stage and should be uploaded as a separate file during online submission labelled as File Type 'Abstract Figure'. Please also ensure that you include the figure legend in the main article file. All Abstract Figures should be created using BioRender. Authors should use The Journal's premium BioRender account to export high-resolution images. Details on how to use and access the premium account are included as part of this email.

Reviewing Editor:

Methods Details:

General rewriting for transparency is required. Number of animals, inclusion/exclusion. Rationale for using mice that are not fully mature, rationale for not including females etc. Expansion of limitations required.

Clarification of light schedules for each experiment are requested by the reviewers for transparency of the results presented. Macro contents of chow. Cage activity/behaviour, feeding behaviours.

Comments to the authors:

We would like to thank the authors for their submission to The Journal of Physiology. We apologize for the longer than normal review time due to difficulties securing reviewers over the winter break.

Both reviewers saw a lot of potential in this work and commended the complementary experiments, however both reviewers believe that in order for this paper to be of major impact it would require judicious rewriting and clarification of results, including appropriate interpretation of circadian concepts as outlined in their comments.

I have summarized some major concerns below that need addressing before we can reconsider this manuscript for the journal.

-Interpretation of concepts such as Zeitgeber are flawed or poorly justified.

-Time of day in human samples needs to be corrected for time of day in mice.

-The KYN pathway results seem contradictory

-A summary of how the KYN pathway has been tipped towards neurotoxicity or the opposite should be made for each appropriate section.

-Rationale for studying muscle and hippocampus specifically. Why not other brain regions or peripheral tissues such as liver/kidney which may deal with the metabolites eventually.

-Inclusion of investigation into other circadian clock factors may be important for interpretations.

-Justification of phosphorylated vs total Reverb α .

-Consideration of extending timepoints to consider the data as 'circadian'.

-Addressing limitations.

Senior Editor:

Comments to ensure the paper complies with the Statistics Policy:

Please familiarize yourself with The Journal's statistics policy and make certain that your revised manuscript is in compliance. For example, the policy requires that precise p-values be provided with the data rather than symbols indicating $p < 0.05$. Thank you in advance.

Comments to the Author:

Thank you for submitting your manuscript for consideration by The Journal of Physiology. As part of the peer review process, we recruited two Referees with expertise in this field of study. While both Referees note that they find the study relevant and interesting, they also note some major concerns that would need to be addressed including concerns about the presentation and interpretation of the results and overall communication in each section of the paper. We would like to invite you to respond point-by-point to each of the Referee's and Reviewing Editor's concerns with corresponding revisions to the manuscript. We thank you for your interest in The Journal and look forward to seeing the revised manuscript.

Referee #1:

The interaction between circadian rhythms and exercise is topical and one that has a wide range of interest. The research presented in this manuscript examines how acute and long-term exposure of mice to paradigms that result in physical exhaustion affect the expression of the clock gene/protein Nr1d1/Reverb alpha in skeletal muscle and the hippocampus of the brain. A goal is to relate this to kynurenine pathway which is implicated in fatigue and this pathway can promote the production of neurotoxic compounds. This is examined through measuring kynurenine aminotransferase 1 (KAT1) which when elevated reduces the production of neurotoxic compounds and kynurenine-3-monooxygenase (KMO) whose elevated expression is associated with neurotoxic aspects of this pathway. Mice in some studies were also treated with a Reverb alpha agonist, SR9009. The in vivo studies are complemented by examination of human datasets as well as cultured C2C12 myoblasts.

I found the experimental design to present problems for the clarity expression of the findings with seemingly contradictory outcomes from acute and over training conditions, depending on whether brain or muscle examined. Further, the definition of Zeitgeber time (spelt incorrectly page 14, line 362) is incorrect. The onset of the dark phase is ZT12, not ZT0. This makes comparisons over time difficult to assess as there are potentially two Zeitgebers--the LD cycle and the overtraining protocol. Other comparisons such as phosphorylated reverb alpha levels and Nr1d1 mRNA are confusing. Further, the authors never state why they chose the hippocampus to examine. The findings on the KYN pathway are also unclear or at least the presentation of the findings makes it difficult to discern the key trends.

Comparisons to human data need to be adjusted for diurnal vs nocturnal animals.

Referee #2:

The current study by da Rocha et al. is on a topic of relevance and general interest to the readers of the journal. I would like to express my appreciation for the authors to perform these kind of experiments which are not trivial and provide unique temporal insights into physiology. The authors used reasonable methods, and I appreciate that the authors assessed protein as well as transcript levels of the most important "players" involved instead of less informative but nowadays more commonly reported transcript levels only. I also appreciate the summarizing sketches from Biorender at the end of most figures as well as the conclusive sketch figure in the very end. However, I also have several comments on most sections of the manuscript that I would like to see addressed by the authors in a revision.

Abstract:

The abstract lacks an explanation about the type of exhaustive exercise (EE) applied in this study.

Introduction:

The authors do not convincingly elaborate on the societal or clinical relevance of EE beyond athletic populations. And it seems questionable whether targeting the Reverb-alpha and/or kynurenine pathway is worth pursuing to avoid maladaptation to EE in athletes.

In light of "Time- and exercise-dependent gene regulation in human skeletal muscle" (Zamboni et al. 2003) and other available studies in mice, EE through resistance training and its impact on clock gene expression in skeletal muscle could also be discussed in comparison to the endurance training-induced EE of the studies referred to in the introduction and of the present study. Potential differences between resistance- and endurance-induced EE could be shortly discussed, is there evidence to suggest that the KYN pathway is triggered for both types?

Line 73-85: Are these findings all based on experiments in the murine hippocampus or also other tissues/cells? Could the authors clarify this aspect for the respective studies?

Methods:

Experiment 2 and 4 are described in the main text, but experiment 3 is only named as such in one of the figures, could you clearly declare what part of the study is experiment 3 for orientation.

Could sample sizes of mice for each experiment also be stated in the main text in brackets?

I suspect that the rationale of using downhill running for the ILT test was to put more emphasis on eccentric muscle contractions to increase muscle damage, correct? This could shortly be stated and supported with a reference.

I don't understand the purpose of the Rotarod test, could you explain it in a bit more detail in the main text?

Line 165: "After 48 hours of the ILT test, mice were weighed, and the exercise session was initiated between 6-7 pm" What was the rationale to perform this test at the beginning of the dark/active phase? Would you expect similar results in your study if EE was performed in the late active phase? For example Ezagouri et al. (10.1016/j.cmet.2019.03.012) found higher endurance exercise capacity for mice in the late active phase, and Sato et al. (10.1016/j.cmet.2021.12.016) found that the maintenance of inter-tissue metabostasis is specified by exercise timing in mice.

Is it known how SR9009 treatment in vivo mechanistically activates Reverb-alpha?

I wonder why the authors did not continue to collect samples at least 24 hours after the acute EXH or the chronic OTR intervention. This would have allowed for more justified conclusions on 24-hour rhythmicity. In this context, I advise the authors to refrain from using the term "circadian rhythm" whenever they derive their conclusion based on 18 hours of sampling and additionally considering that these mice were kept under entrained rhythmic conditions of light/dark, so that an endogenously driven circadian rhythm was not tested. The authors should instead consider using the term "diurnal rhythm" or "diurnal oscillation". An example would be Line 463: "acute EE reduced REV-ERB- α levels in the gastrocnemius and abolished its circadian oscillation". Therefore, I advise the authors to reevaluate their entire manuscript for more careful phrasing.

Table 2 can go into the Supplement in my view.

Which statistical test was used to test the presence or absence of diurnal rhythmicity in respective outcomes?

Discussion:

60 min following an intense 15 min exercise bout (at 80% VO₂max) on a bicycle ergometer, Small et al. (<https://doi.org/10.1113/JP280428>) recently found that Per1 and Per2 were increased and Nr1d1 was decreased in human skeletal muscle 1-h post-exercise. They also investigated muscle contractions and Ca²⁺ signaling as mechanisms behind these effects. The authors could discuss their findings in light of that study.

Line 362: "Zeitberg" should be replaced by "Zeitgeber".

Line 517: "Protein peaks coincide with the lowest transcriptional levels due to their regulatory oscillation, creating a negative feedback loop that interacts with other clock genes."? Could you point the reader to the respective evidence for this statement?

Line 526: "Another complicating factor is that mice have an inverted wake-rest cycle compared to humans, affecting REV-ERB- α 's circadian rhythm". In this regard, Gutierrez-Monreal et al. (<https://doi.org/10.1002/oby.22826>) compared peak times of expression of a larger composite of clock genes in mouse skeletal muscle and human skeletal muscle. For Reverb-alpha, at least on the mRNA level, they show a consistent peak time at the beginning of the respective rest phase.

Line 567: "Frank et al. (39) found that 68% of elite athletes (Olympic level) had experienced at least one episode of depression." If this study does not link depression to EE, I find this reference misleading, as depression in athletes (as for any other population at risk) can be very multifactorial.

The authors do not address any limitations of their study. For example, as no other exercise intensities have been tested alongside EE in comparison to the sedentary group, we have no idea if some of the findings would also occur for less intense exercise. Also, the tested muscle was the fast-twitch gastrocnemius, would we expect another response in a more slow-twitch muscle like the soleus?

END OF COMMENTS

8th April 2025**Dr. Kim E. Barrett****Editor-in-chief:** The Journal of Physiology

First, we would like to express our sincere gratitude for your valuable feedback and for the opportunity to revise our manuscript, entitled “EXHAUSTIVE EXERCISE ABOLISHES REV-ERB- α CIRCADIAN RHYTHM AND SHIFTS THE KYNURENINE PATHWAY TO A NEUROTOXIC PROFILE IN MICE” (JP-RP-2024-288290). We greatly appreciate the reviewers’ insightful comments, which have significantly contributed to enhancing the clarity and scientific rigor of our work. We are confident that the revisions have substantially improved the manuscript, and we hope our efforts are reflected in both our detailed responses and the revised version. Below, we provide a point-by-point response to the comments from the editors and reviewers, outlining how each concern has been addressed. All changes have been clearly highlighted in the updated manuscript. Please do not hesitate to reach out if further clarification is needed.

Sincerely,

Alisson Luiz da Rocha, Ph.D.

Adelino Sanchez Ramos da Silva, Ph.D.

Reviewing Editor

1. “General rewriting for transparency is required. Number of animals, inclusion/exclusion. Rationale for using mice that are not fully mature, rationale for not including females etc. Expansion of limitations required.”

Answer: We agree with your commentary. Therefore, we have updated the revised manuscript by incorporating this information between lines 146–166 and 716–729.

2. “Clarification of light schedules for each experiment are requested by the reviewers for transparency of the results presented. Macro contents of chow. Cage activity/behaviour, feeding behaviours.”

Answer: We appreciate this concern and have updated the manuscript to include details on Zeitgeber time and the macronutrient composition of the chow. Regarding feeding behavior, we previously published an article addressing this aspect in the context of the overtraining protocol (Pereira *et al.*, 2015). Unfortunately, we do not have data on cage activity for these mice. However, to address the concern about activity phases raised by Referee #2, we have conducted new experiments and included the findings in Response Letter_Figure 1 (unpublished data).

Response letter_Figure 1. Total locomotor activity of mice over 24 hours (n=4). Mice were first adapted to the cage for 24 hours. After this period, the locomotor activity was recorded for an additional 24 hours. The animals were maintained on an inverted wake/rest cycle (lights off: 6 AM/lights on: 6 PM). CTL^{WT}: Wild-type sedentary C57bl/6 mice.

3. “I have summarized some major concerns below that need addressing before we can reconsider this manuscript for the journal.”

a) *Interpretation of concepts such as Zeitgeber are flawed or poorly justified.*

Answer: We agree with this comment. The Zeitgeber time previously referenced in the manuscript specifically refers to the light/dark cycle. To enhance clarity, we have revised the manuscript to explicitly define ZT12 as the onset of the dark phase and ZT0 as the onset of the light phase.

b) *Time of day in human samples needs to be corrected for time of day in mice.*

Answer: We agree with this comment and have updated the revised manuscript (lines 629–630) to clarify this phenomenon. Additionally, using a public dataset, we developed and included a figure illustrating this aspect in Figures 4c and 4d in the revised version of the manuscript.

c) *The KYN pathway results seem contradictory*

Answer: We understand this concern. In skeletal muscle, both acute and chronic exhaustive exercise protocols produced similar responses, characterized by a reduction in *KAT1/Kyat1* levels and an upregulation of *Kmo*, indicating a shift toward the neurotoxic branch of the kynurenine pathway. Conversely, treatment with a REV-ERB- α agonist, known to enhance exercise tolerance (Woldt *et al.*, 2013), had the opposite effect, increasing *KAT1/Kyat1* levels.

In the acute protocol, skeletal muscle exhibited higher KYNA levels and lower KYN levels, suggesting increased conversion of KYN to KYNA. However, the observed decrease in KAT1, the enzyme responsible for this conversion, appears contradictory. This paradox may be explained by regulatory mechanisms such as negative feedback loops or enzyme degradation due to excessive activation, leading to a compensatory reduction in KAT1 levels. In the chronic protocol, the reduction in KAT1 was not accompanied by increased KYNA levels. Furthermore, since the mice were analyzed 48 hours post-exercise, the tissue was in a basal state, likely reflecting an adaptation phase. This suggests that the reduction in KAT1 is a maladaptive response, impairing the muscle’s ability to convert KYN into KYNA and diminishing its neuroprotective potential (Agudelo *et al.*, 2014).

In the hippocampus, both acute and chronic exhaustive exercise protocols led to the downregulation of enzymes responsible for KYN metabolism. Specifically, in the acute condition, KYN bioavailability increased, potentially triggering a compensatory downregulation of its metabolizing enzymes as a protective mechanism. This regulation may prevent excessive accumulation of KYNA or other neuroactive kynurenine metabolites, such as 3-HK and quinolinic acid, which can be neurotoxic. Maintaining a precise balance of kynurenine metabolites is essential for neural health, as disruptions in this balance are associated with neurological disorders—elevated KYNA levels are linked to schizophrenia, while increased 3-HK and quinolinic acid levels are implicated in depression, Huntington's disease, and Alzheimer's disease (Pathak *et al.*, 2024).

d) *A summary of how the KYN pathway has been tipped towards neurotoxicity or the opposite should be made for each appropriate section.*

Answer: We agree with this suggestion. To enhance clarity, we have added summarizing paragraphs at the end of each relevant section in the Discussion, highlighting the key findings on how the KYN pathway is shifted toward neurotoxicity or neuroprotection (lines 638–643 and 684–693).

e) *Rationale for studying muscle and hippocampus specifically. Why not other brain regions or peripheral tissues such as liver/kidney which may deal with the metabolites eventually.*

Answer: We agree with this comment. To address this, we have updated the Introduction by adding a sentence highlighting the relevance of the hippocampus in our study (lines 123–126).

f) *Inclusion of investigation into other circadian clock factors may be important for interpretations.*

Answer: We understand this concern. Assessing additional circadian clock factors could indeed provide further insights into the effects of exhaustive exercise on circadian rhythm regulation. However, our initial findings guided us toward a more focused investigation of the relationship between REV-ERB- α and the kynurenine pathway. While we acknowledge the relevance of analyzing clock gene expression following exhaustive exercise, our primary objective was to elucidate new mechanistic interactions between these two regulatory components. To achieve this, we employed genetic manipulation and pharmacological

treatment strategies for a more in-depth exploration. Nonetheless, we recognize this as a limitation of our study and have addressed it in the Limitations and Future Perspectives section of the revised manuscript.

g) Justification of phosphorylated vs total Reverbalpha.

Answer: We understand this concern. In this study, we present different forms of REV-ERB- α expression: total, phosphorylated, and their ratio. Post-translational modifications (PTMs), including phosphorylation, acetylation, methylation, among several others, play a critical role in modulating various functional properties of proteins, such as stability, activity, localization, and interactions with other biomolecules. (Ramazi & Zahiri, 2021). To more accurately assess PTMs modifications, normalizing phosphorylated levels to total protein provides a clearer perspective. This is particularly evident in skeletal muscle samples from Experiment 1, where total REV-ERB- α is reduced, but phosphorylated levels remain unchanged. This dynamic is more clearly observed in the ratio, which may reflect a mechanism to enhance protein stability, as discussed in the revised manuscript (lines 564–576).

h) Consideration of extending timepoints to consider the data as 'circadian'.

Answer: We agree with this suggestion. Unfortunately, we were unable to conduct analyses at extended time points. Therefore, as recommended by Referee #2, we have updated the manuscript to use the term diurnal rhythm instead of circadian rhythm.

i) Addressing limitations.

Answer: We appreciate this comment. Accordingly, we have revised the manuscript to explicitly address the study's limitations (lines 716–729).

Senior Editor

1. *“Comments to ensure the paper complies with the Statistics Policy: Please familiarize yourself with The Journal’s statistics policy and make certain that your revised manuscript is in compliance. For example, the policy requires that precise p-values be provided with the data rather than symbols indicating $p < 0.05$. Thank you in advance.”*

Answer: We appreciate this feedback. Accordingly, we have revised the manuscript to include exact p-values for all results presented, ensuring compliance with the journal’s statistics policy.

Referee #1

“The interaction between circadian rhythms and exercise is topical and one that has a wide range of interest. The research presented in this manuscript examines how acute and long-term exposure of mice to paradigms that result in physical exhaustion affect the expression of the clock gene/protein Nr1d1/Reverb alpha in skeletal muscle and the hippocampus of the brain. A goal is to relate this to kynurenine pathway which is implicated in fatigue and this pathway can promote the production of neurotoxic compounds. This is examined through measuring kynurenine aminotransferase 1 (KAT1) which when elevated reduces the production of neurotoxic compounds and kynurenine-3-monooxygenase (KMO) whose elevated expression is associated with neurotoxic aspects of this pathway. Mice in some studies were also treated with a Reverb alpha agonist, SR9009. The in vivo studies are complemented by examination of human datasets as well as cultured C2C12 myoblasts.”

1. *“I found the experimental design to present problems for the clarity expression of the findings with seemingly contradictory outcomes from acute and over training conditions, depending on whether brain or muscle examined”.*

Answer: We understand this concern and appreciate the opportunity to clarify. In skeletal muscle, both acute and chronic protocols yielded similar responses at the mRNA and protein levels, with downregulation of total REV-ERB- α and KAT1, alongside an increased phospho/total ratio for REV-ERB- α . This pattern suggests a maladaptive response initiated by the first bout of exhaustive exercise, which persisted after the 8-week overtraining protocol. Skeletal muscle is directly impacted by downhill running, which can induce damage. Without sufficient recovery, this damage accumulates, reinforcing the harmful stimulus.

In contrast, the central nervous system (CNS) is indirectly affected by factors released during exhaustive exercise. The acute response may represent a protective mechanism aimed at stabilizing the CNS environment, whereas repeated exposure during the chronic protocol could drive metabolic adaptations in CNS cells. Additionally, tissue collection in the chronic protocol occurred 48 hours after the last exercise session, likely capturing an adaptation phase rather than an immediate response. To

address this concern, we have updated the Discussion section to explicitly clarify these aspects (lines 648–650).

2. *“Further, the definition of Zeitgeber time (spelt incorrectly page 14, line 362) is incorrect. The onset of the dark phase is ZT12, not ZT0. This makes comparisons over time difficult to assess as there are potentially two Zeitgebers--the LD cycle and the overtraining protocol”.*

Answer: We agree with this comment. The Zeitgeber time referenced in the manuscript specifically refers to the light/dark cycle. Accordingly, we have revised the manuscript to explicitly define ZT12 as the onset of the dark phase and ZT0 as the onset of the light phase to ensure clarity.

3. *“Other comparisons such as phosphorylated reverb alpha levels and Nr1d1 mRNA are confusing”.*

Answer: We understand this concern. The correlation between total and phosphorylated REV-ERB- α in the CTL group within the gastrocnemius muscle (Figure 4a) reflects the expected relationship under baseline conditions. These findings suggest a potential link between circadian oscillations and the post-translational modification of REV-ERB- α . To clarify this, we have revised the corresponding section of the Discussion (lines 566–567). Additionally, we have revised the paragraphs discussing the correlations between total REV-ERB- α , *Nr1d1*, and KMO (mRNA and protein levels) to improve clarity and conciseness in the revised manuscript (lines 564–588).

4. *Further, the authors never state why they chose the hippocampus to examine*

Answer: We agree with this comment. To address this, we have updated the Introduction to include a sentence highlighting the relevance of the hippocampus in our study (lines 123–126).

5. *“The findings on the KYN pathway are also unclear or at least the presentation of the findings makes it difficult to discern the key trends”.*

Answer: We understand this concern. To enhance clarity, we have reorganized the manuscript and included summary descriptions of the key findings at the end of each relevant section (lines 638–643 and 684–693).

6. *“Comparisons to human data need to be adjusted for diurnal vs nocturnal animals”.*

Answer: We agree with this comment. Accordingly, we have updated the revised manuscript (lines 629–630) to highlight this distinction. Additionally, using a public dataset, we developed and included a figure illustrating this aspect in Figures 4c and 4d.

Referee #2

“The current study by da Rocha et al. is on a topic of relevance and general interest to the readers of the journal. I would like to express my appreciation for the authors to perform these kind of experiments which are not trivial and provide unique temporal insights into physiology. The authors used reasonable methods, and I appreciate that the authors assessed protein as well as transcript levels of the most important “players” involved instead of less informative but nowadays more commonly reported transcript levels only. I also appreciate the summarizing sketches from Biorender at the end of most figures as well as the conclusive sketch figure in the very end. However, I also have several comments on most sections of the manuscript that I would like to see addressed by the authors in a revision”.

1. *“Introduction: The authors do not convincingly elaborate on the societal or clinical relevance of EE beyond athletic populations. And it seems questionable whether targeting the Reverb-alpha and/or kynurenine pathway is worth pursuing to avoid maladaptation to EE in athletes”.*

Answer: We understand this concern. While excessive exercise is widely recognized in elite athletes, where training is performance-driven, it also occurs in the general population, often as a form of behavioral addiction. Chhabra *et al.* (Chhabra *et al.*, 2024) reported a 9% prevalence of exercise addiction across 15 countries, with no significant gender differences, while athletes exhibited twice the risk of developing this condition. In both athletes and the general population, excessive exercise has been linked to behavioral disturbances, underscoring the importance of understanding its underlying mechanisms. In this context, our findings suggest that *Nr1d1* dysregulation may alter the kynurenine pathway, potentially affecting the central nervous system and contributing to cognitive and behavioral disturbances. This novel insight identifies a potential mechanistic link that could serve as a therapeutic target for future research. To emphasize this perspective, we have added a sentence to the Introduction (lines 106–108).

2. *Introduction: In light of "Time- and exercise-dependent gene regulation in human skeletal muscle" (Zambon et al. 2003) and other available studies in mice, EE through resistance training and its impact on clock gene expression in skeletal muscle could also be discussed in comparison to the endurance training-induced EE of the studies referred to in the introduction and of the present study. Potential differences between resistance- and endurance-induced EE could be shortly discussed, is there evidence to suggest that the KYN pathway is triggered for both types?"*

Answer: We agree with this suggestion. Accordingly, we have added a sentence addressing this issue in the revised manuscript (lines 602–604).

3. *Introduction: Line 73-85: Are these findings all based on experiments in the murine hippocampus or also other tissues/cells? Could the authors clarify this aspect for the respective studies?"*

Answer: We agree with this comment. Accordingly, we have clarified this aspect by adding the relevant details to the Introduction (lines 129–131).

4. *Methods: Experiment 2 and 4 are described in the main text, but experiment 3 is only named as such in one of the figures, could you clearly declare what part of the study is experiment 3 for orientation."*

Answer: We agree with this comment. To improve clarity, we have added a sentence at the beginning of line 168 explicitly stating that "In vitro experiments" refers to Experiment 3.

5. *Methods: Could sample sizes of mice for each experiment also be stated in the main text in brackets?"*

Answer: We agree with this comment. Accordingly, we have added the sample sizes in the main text (lines 156–166).

6. *Methods: I don't understand the purpose of the Rotarod test, could you explain it in a bit more detail in the main text?"*

Answer: We understand this concern. To clarify the purpose of the Rotarod test, we have added further details in the main text (lines 208–210).

7. "Methods: I suspect that the rationale of using downhill running for the ILT test was to put more emphasis on eccentric muscle contractions to increase muscle damage, correct? This could shortly be stated and supported with a reference".

Answer: We agree with this comment. Accordingly, we have added a sentence in the revised manuscript (lines 204–206) to clarify the rationale for using downhill running in the ILT test, supported by a reference.

8. "Methods: Line 165: "After 48 hours of the ILT test, mice were weighed, and the exercise session was initiated between 6-7 pm" What was the rationale to perform this test at the beginning of the dark/active phase? Would you expect similar results in your study if EE was performed in the late active phase? For example Ezagouri et al. (10.1016/j.cmet.2019.03.012) found higher endurance exercise capacity for mice in the late active phase, and Sato et al. (10.1016/j.cmet.2021.12.016) found that the maintenance of inter-tissue metabostasis is specified by exercise timing in mice".

Answer: We understand this concern. The timing of exercise indeed influences metabolic responses in skeletal muscle and interorgan crosstalk. However, evidence also suggests that mice exhibit greater spontaneous activity during the early active phase (Dial et al., 2024). Both exhaustive exercise protocols were designed to allow mice to run until fatigue with minimal external stimulation. To further illustrate this activity pattern, we conducted an additional analysis using male wild-type sedentary mice (CTL^{WT}) of the same age as those in the present study (Response letter_Figure 1 – unpublished data). The mice (n = 4) were placed in a locomotor activity cage (ACTITRACK®, Panlab, Barcelona, Spain) for a 24-hour adaptation period, followed by a 24-hour recording session. The results showed a peak in locomotor activity at the beginning of the active phase.

Response letter_Figure 1. Total locomotor activity of mice over 24 hours (n=4). Mice were first adapted to the cage for 24 hours. After this period, the locomotor activity was recorded for an additional 24 hours. The animals were maintained on an inverted wake/rest cycle (lights off: 6 AM/lights on: 6 PM). CTL^{WT}: Wild-type sedentary C57bl/6 mice.

9. *“Methods: Is it known how SR9009 treatment in vivo mechanistically activates Rev-erb- α ?”*

Answer: Unfortunately, the exact mechanism by which SR9009 regulates REV-ERB- α remains unclear. However, Yin *et al.* (Yin *et al.*, 2007) demonstrated that the endogenous ligand heme enhances REV-ERB- α stability and promotes the recruitment of NCOR and HDAC3, thereby facilitating its nuclear repression activity. The same study identified Histidine 602 (H602) as a crucial binding site for heme, essential for REV-ERB- α function (Yin *et al.*, 2007). Additionally, REV-ERB- α can be phosphorylated at serines 55 and 59, which increases its stability (Yin *et al.*, 2006). Given heme’s role as an endogenous ligand for REV-ERB- α , other studies have shown that SR9009 enhances REV-ERB- α activity by reducing the expression of its circadian targets *Bmal1* and *Clock* (Sheng *et al.*, 2023) and by increasing the expression of REV-ERB- β downstream target genes, such as *Ccna2* and *Sez6* (Shimozaki, 2022). Collectively, these findings suggest that SR9009 enhances REV-ERB- α activity like heme; however, the precise molecular mechanism remains unknown.

10. *“Methods: I wonder why the authors did not continue to collect samples at least 24 hours after the acute EXH or the chronic OTR intervention. This would have allowed for more justified conclusions on 24-hour rhythmicity. In this context, I advise the authors to refrain from using the term “circadian rhythm” whenever they derive their conclusion based on 18 hours of sampling and additionally*

considering that these mice were kept under entrained rhythmic conditions of light/dark, so that an endogenously driven circadian rhythm was not tested. The authors should instead consider using the term "diurnal rhythm" or "diurnal oscillation". An example would be Line 463: "acute EE reduced REV-ERB- α levels in the gastrocnemius and abolished its circadian oscillation". Therefore, I advise the authors to reevaluate their entire manuscript for more careful phrasing".

Answer: We agree with this suggestion. Accordingly, we have revised the manuscript to replace the term circadian rhythm with diurnal rhythm where appropriate.

11. *"Methods: Table 2 can go into the Supplement in my view".*

Answer: We understand your concern. Due to the journal's policy, we are unable to publish supplementary files containing graphs or tables. Therefore, we have updated the entire manuscript to include all relevant data cited in the main text. Additionally, we have incorporated the primer design information directly into the text, between lines 255–266.

12. *"Methods: Which statistical test was used to test the presence or absence of diurnal rhythmicity in respective outcomes?"*

Answer: We understand this concern. To address it, we performed additional analyses using the MetaCycle package with the JTK_CYCLE algorithm to evaluate the amplitude and rhythmicity of *Nr1d1* and REV-ERB- α in skeletal muscle and the hippocampus. However, due to the limited number of time points (0, 6, 12, and 18 hours for the acute exercise protocol, and ZT0, 6, 12, and 18 for the chronic exercise protocol), the analysis did not yield statistically significant rhythmicity values. Nonetheless, using a public dataset, we confirmed the rhythmicity of *Nr1d1* in skeletal muscle (Response Letter_Figure 2). Additionally, we have incorporated the amplitude analysis into the revised manuscript (Figures 3i, 3j, 6r, and 6s).

DATASET GSE197726						
Nr1d1 mRNA levels (normalized) - skeletal muscle						
Mice ID	ZT0	ZT4	ZT8	ZT12	ZT16	ZT20
1	9.71927	15.9962	12.2513	5.61927	2.9573	3.84632
2	5.77992	10.6989	10.0793	4.96003	3.23547	3.44111
3	7.35072	14.9776	11.3416	4.91743	3.41456	4.34583
MetaCycle output						
CyclID	JTK_pvalue	JTK_BH.Q	JTK_period			
1	0.03055556	0.03055556	24			
2	0.03055556	0.03055556	24			
3	0.03055556	0.03055556	24			

Response Letter_Figure 2. MetaCycle output using the JTK_CYCLE algorithm to analyze the rhythmicity of *Nr1d1* in the skeletal muscle of mice from the public dataset GSE197726; n=3; animal model= C57BL/6J wildtype mice from 8 to 12 weeks; 12hrs light /12hrs dark with *ad libitum* access to water and food.

13. *Discussion:* 60 min following an intense 15 min exercise bout (at 80% VO₂max) on a bicycle ergometer, Small et al. (<https://doi.org/10.1113/JP280428>) recently found that *Per1* and *Per2* were increased and *Nr1d1* was decreased in human skeletal muscle 1-h post-exercise. They also investigated muscle contractions and Ca²⁺ signaling as mechanisms behind these effects. The authors could discuss their findings in light of that study”.

Answer: We agree with this suggestion. Accordingly, we have added a paragraph discussing these findings in the revised manuscript (lines 558–563).

14. *Discussion:* Line 362: ‘Zeitberg’ should be replaced by ‘Zeitgeber’”.

Answer: We appreciate this comment. Accordingly, we have corrected this typo and ensured consistency throughout the manuscript.

15. *Discussion:* Line 517: “Protein peaks coincide with the lowest transcriptional levels due to their regulatory oscillation, creating a negative feedback loop that interacts with other clock genes.”? Could you point the reader to the respective evidence for this statement?”

Answer: We appreciate this comment. Accordingly, we have added a reference to support the statement regarding the negative feedback loop involving clock genes and included an illustration in Figure 4b of the revised manuscript to highlight this regulatory pattern.

16. *“Discussion: Line 526: “Another complicating factor is that mice have an inverted wake-rest cycle compared to humans, affecting REV-ERB- α ’s circadian rhythm”. In this regard, Gutierrez-Monreal et al. (<https://doi.org/10.1002/oby.22826>) compared peak times of expression of a larger composite of clock genes in mouse skeletal muscle and human skeletal muscle. For Revb- α , at least on the mRNA level, they show a consistent peak time at the beginning of the respective rest phase”.*

Answer: We appreciate this comment. Using public datasets, we generated a figure illustrating the oscillation of *Nr1d1* throughout the day, comparing humans and mice (Figures 4c and 4d). Accordingly, we have updated the Discussion to reflect this information (lines 629–630).

17. *“Discussion: Line 567: “Frank et al. (39) found that 68% of elite athletes (Olympic level) had experienced at least one episode of depression.” If this study does not link depression to EE, I find this reference misleading, as depression in athletes (as for any other population at risk) can be very multifactorial”.*

Answer: We agree with this suggestion. Accordingly, we have revised the sentence to more accurately reflect the original idea (lines 679–683).

18. *“Discussion: The authors do not address any limitations of their study. For example, as no other exercise intensities have been tested alongside EE in comparison to the sedentary group, we have no idea if some of the findings would also occur for less intense exercise. Also, the tested muscle was the fast-twitch gastrocnemius, would we expect another response in a more slow-twitch muscle like the soleus?”*

Answer: We appreciate this comment. Accordingly, we have revised the manuscript to explicitly address the study’s limitations (lines 716–729).

REFERENCES

- Agudelo LZ, Femenía T, Orhan F, Porsmyr-Palmertz M, Goiny M, Martinez-Redondo V, Correia JC, Izadi M, Bhat M & Schuppe-Koistinen I. (2014). Skeletal muscle PGC-1 α 1 modulates kynurenine metabolism and mediates resilience to stress-induced depression. *Cell* **159**, 33-45.
- Chhabra B, Granzio U, Griffiths MD, Zandonai T, Landolfi E, Solmi M, Zou LY, Yang PY, Lichtenstein MB, Stoll O, Akimoto T, Cantù-Berrueto A, Larios A, Egorov AY, Marcos RD, Alpay M, Nazligul MD, Yildirim M, Trott M, Portman RM & Szabo A. (2024). Prevalence of the Risk of Exercise Addiction Based on a New Classification: A Cross-Sectional Study in 15 Countries. *Int J Ment Health Ad.*
- Dial MB, Malek EM, Neblina GA, Cooper AR, Vaslieva NI, Frommer R, Girgis M, Dawn B & McGinnis GR. (2024). Effects of time-restricted exercise on activity rhythms and exercise-induced adaptations in the heart. *Sci Rep* **14**, 146.
- Pathak S, Nadar R, Kim S, Liu K, Govindarajulu M, Cook P, Watts Alexander CS, Dhanasekaran M & Moore T. (2024). The Influence of Kynurenine Metabolites on Neurodegenerative Pathologies. *Int J Mol Sci* **25**.
- Pereira BC, da Rocha AL, Pauli JR, Ropelle ER, de Souza CT, Cintra DE, Sant'Ana MR & da Silva AS. (2015). Excessive eccentric exercise leads to transitory hypothalamic inflammation, which may contribute to the low body weight gain and food intake in overtrained mice. *Neuroscience* **311**, 231-242.

- Ramazi S & Zahiri J. (2021). Posttranslational modifications in proteins: resources, tools and prediction methods. *Database (Oxford)* **2021**.
- Sheng M, Chen X, Yu Y, Wu Q, Kou J & Chen G. (2023). Rev-erbalpha agonist SR9009 protects against cerebral ischemic injury through mechanisms involving Nrf2 pathway. *Front Pharmacol* **14**, 1102567.
- Shimozaki K. (2022). REV-ERB Agonist SR9009 Regulates the Proliferation and Neurite Outgrowth/Suppression of Cultured Rat Adult Hippocampal Neural Stem/Progenitor Cells in a Concentration-Dependent Manner. *Cell Mol Neurobiol* **42**, 1765-1776.
- Woldt E, Sebti Y, Solt LA, Duhem C, Lancel S, Eeckhoutte J, Hesselink MK, Paquet C, Delhay S & Shin Y. (2013). Rev-erb- α modulates skeletal muscle oxidative capacity by regulating mitochondrial biogenesis and autophagy. *Nature medicine* **19**, 1039-1046.
- Yin L, Wang J, Klein PS & Lazar MA. (2006). Nuclear receptor Rev-erbalpha is a critical lithium-sensitive component of the circadian clock. *Science* **311**, 1002-1005.
- Yin L, Wu N, Curtin JC, Qatanani M, Szwegold NR, Reid RA, Waitt GM, Parks DJ, Pearce KH, Wisely GB & Lazar MA. (2007). Rev-erbalpha, a heme sensor that coordinates metabolic and circadian pathways. *Science* **318**, 1786-1789.

Dear Dr da Rocha,

Re: JP-RP-2025-288290R1 "EXHAUSTIVE EXERCISE ABOLISHES REV-ERB- α CIRCADIAN RHYTHM AND SHIFTS THE KYNURENINE PATHWAY TO A NEUROTOXIC PROFILE IN MICE" by Alisson Luiz da Rocha, Ana Paula Pinto, Ivo Vieira de Sousa Neto, Vitor Rosetto Munoz, Bruno Brieda Marafon, Lilian Esleine Costa Mendes da Silva, José Rodrigo Pauli, Dennys Esper Cintra, Eduardo R Ropelle, Fernando Simabuco, Leandro P de Moura, Ellen Cristini de Freitas, Donato Americo Rivas, and Adelino Sanchez Ramos da Silva Silva

Thank you for submitting your manuscript to The Journal of Physiology. It has been assessed by a Reviewing Editor and by 2 expert referees and we are pleased to tell you that it is acceptable for publication following satisfactory revision.

REVISION CHECKLIST:

Please upload two versions of your manuscript text: one with all relevant changes highlighted and one clean version with no changes tracked. The manuscript file should include all tables and figure legends, but each figure/graph should be uploaded as separate, high-resolution files. The journal is now integrated with Wiley's Image Checking service. For further details, see: <https://www.wiley.com/en-us/network/publishing/research-publishing/trending-stories/upholding-image-integrity-wileys->

image-screening-service

We look forward to receiving your revised submission.

Yours sincerely,

Karyn Hamilton
Senior Editor
The Journal of Physiology

EDITOR COMMENTS

Reviewing Editor:

Thankyou for your detailed rebuttal in response to our expert reviewers comments which were well received and have considerably improved the readability, quality and direction of the manuscript.

However, further concerns were made about the lack of behavioural/cognitive assessments made to assess hippocampal/neurological function. In the manuscripts current state, it is difficult to assess the translatability of the model to humans, or for the reader to establish whether increases in KYNA seen in this exercise model are similar to pathological levels observed in neurological disorder mouse models. Assessing this would greatly improve the impact of this manuscript and translatability. Assessment of liver or discussion of its role in this model given its importance in murine energy metabolism and inflammation would also be of interest.

The reviewers also had further minor suggestions to improve clarity and quality of the text.

Senior Editor:

Thank you for your carefully revised manuscript. The Referees were largely satisfied with changes you made in response to their feedback. A few suggestions remain. At this point, we'd like to Provisionally Accept your manuscript pending these final revisions, as summarized by the Reviewing Editor. Thank you and we look forward to seeing your revised work!

Please check the statistics policy one more time. I think I noted a couple of remaining figures where individual data points are not represented on the histograms.

REFEREE COMMENTS

Referee #1:

The authors have done a good job revising their manuscript. The revisions are thorough and the quality of writing very good. I think it is much easier to follow the rationale for selecting the tissues that they did. I was a little surprised given how the hippocampus is a target for the molecules in question that the authors had not attempted some cognitive tests to investigate functional changes in hippocampal function.

Referee #2:

I would like to compliment the authors for substantially improving the manuscript with a lot of edits that had to be made. The authors have addressed most of my concerns and comments in a sufficient manner. Only the following minor comments remain from my side:

Line 316 "Specifically, the JTK_CYCLE algorithm was applied to assess circadian oscillations in gene and protein expression over time (Hughes et al., 2010)."

Please replace circadian by diurnal again.

Line 350 "The EXH group exhibited a reduced amplitude of REV-ERB- α expression in gastrocnemius samples compared to the CTL group ($p=0.001$; Figure 3i)."

I cannot follow the authors here, as they state in the rebuttal letter, that "the analysis did not yield statistically significant rhythmicity values". If no significant rhythmicity was found in the present data, further testing for amplitude (in a rhythmicity context) is not justified. I would advise to refrain from using the word "amplitude" as well. Similarly, I don't think the summary sketch on these findings in the abstract is justified, as it suggests continuous (beyond 24 hours) sampling, which was not done. Instead, the authors could perform an equivalent to the 2-way ANOVA and test for an interaction effect, which could hint towards an amplitude effect if tissues were sampled for longer. If true, this could be discussed.

In some cases, "circadian rhythm" has now been changed to "daily rhythm" instead of as stated in the rebuttal letter "diurnal rhythm". Please make sure to use "diurnal rhythm" throughout.

I am sorry to say that I don't think that adding Figure 4 has much additional value, the correlation is not very convincing. And particularly for 4c and d, there is no need to show these data again, as Nr1d1 mRNA is indeed well known to oscillate in a circadian manner, which can be referred to. However, it does not change the fact that it was not confirmable in the present data due to lack of samples. 4b is the exception of having additional value, although such delay between mRNA and protein levels was to be expected based on Robles et al. (2014; 10.1371/journal.pgen.1004047), who showed that "the oscillations of almost half of the cycling liver proteome were delayed by more than six hours with respect to the corresponding, rhythmic mRNA". Overall, I think the new section 3.2.5 can mostly be removed. With my initial comment, I only meant to rebut your initial point "Another complicating factor is that mice have an inverted wake-rest cycle compared to humans, affecting REV-ERB- α 's circadian rhythm." The authors could simply address this by referring to Gutierrez-Monreal et al. (<https://doi.org/10.1002/oby.22826>), who compared peak times of expression of a larger composite of clock genes in mouse skeletal muscle and human skeletal muscle, and found for Rev-erb- α as well as other clock genes that they show mostly consistent peak times, when normalizing the time points to the respective active/rest phase.

Line 464-466 and Figure 6r and s: Again, I don't think without significant rhythmicity as well as too short and unfrequent tissue sampling, the authors cannot test for amplitude here. Please also check the discussion for the misuse of the term "amplitude", if meant as now illustrated in the abstract figure with the smaller oscillating wave.

Line 487: It says Figure 4e, but should say Figure 7e.

Line 626-630: The authors could reconsider the relevance of this statement in light of the above mentioned Robles et al. paper. In short, there is a transcript to protein translation delay (for most proteins about 4-6 hours), this is likely to explain the inverse relationship in these data.

END OF COMMENTS

20th May 2025**Dr. Kim E. Barrett****Editor-in-chief:** The Journal of Physiology

We want to extend our sincere gratitude to the Reviewing Editor, Senior Editor, and Referees for their thoughtful and constructive feedback during this second round of revisions to our manuscript entitled, "EXHAUSTIVE EXERCISE ABOLISHES REV-ERB- α CIRCADIAN RHYTHM AND SHIFTS THE KYNURENINE PATHWAY TO A NEUROTOXIC PROFILE IN MICE" (JP-RP-2025-288290R2). The reviewers' insightful comments have once again contributed significantly to enhancing the clarity, scientific rigor, and translational relevance of our work. We have carefully addressed all remaining concerns and believe the additional revisions have further strengthened the manuscript. We hope these improvements are evident in our detailed responses and the revised version. Below, we provide a point-by-point response to the comments the editorial team and reviewers raised, indicating how they have been addressed in the updated manuscript. All modifications have been highlighted. Please do not hesitate to contact us if you require any further clarification.

Sincerely,

Alisson Luiz da Rocha, Ph.D.

Adelino Sanchez Ramos da Silva, Ph.D.

Reviewing Editor

1. *"Thank you for your detailed rebuttal in response to our expert reviewers comments which were well received and have considerably improved the readability, quality and direction of the manuscript. However, further concerns were made about the lack of behavioral/cognitive assessments made to assess hippocampal/neurological function."*

Answer: We understand your concern and acknowledge the importance of behavioral and cognitive assessments in the context of excessive training. Although we did not perform behavioral tests in our experimental animals, we have added a statement in the Discussion (lines 718-720) noting that previous studies have reported cognitive and behavioral impairments induced by excessive training in mice. Nonetheless, this study limitation has also been addressed in the corresponding section of the manuscript.

2. *"In the manuscripts current state, it is difficult to assess the translatability of the model to humans, or for the reader to establish whether increases in KYNA seen in this exercise model are similar to pathological levels observed in neurological disorder mouse models. Assessing this would greatly improve the impact of this manuscript and translatability."*

Answer: We agree with your comment. Accordingly, we have inserted a paragraph addressing this topic between lines 695 and 708.

3. *Assessment of liver or discussion of its role in this model given its importance in murine energy metabolism and inflammation would also be of interest."*

Answer: We agree with your commentary. Therefore, we inserted a paragraph addressing this topic between lines 674 and 684.

Senior Editor

1. *"Thank you for your carefully revised manuscript. The Referees were largely satisfied with changes you made in response to their feedback. A few suggestions remain. At this point, we'd like to Provisionally Accept your manuscript pending these final revisions, as summarized by the Reviewing Editor. Thank you and we look forward to seeing your revised work! Please check the statistics policy one more time. I think I noted a couple of remaining figures where individual data points are not represented on the histograms"*

Answer: We agree with your comment. Accordingly, we have updated the remaining histograms to include individual data points.

Referee #1

1. *"The authors have done a good job revising their manuscript. The revisions are thorough and the quality of writing very good. I think it is much easier to follow the rationale for selecting the tissues that they did. I was a little surprised given how the hippocampus is a target for the molecules in question that the authors had not attempted some cognitive tests to investigate functional changes in hippocampal function".*

Answer: We understand your concern. However, our initial findings directed us toward a more focused investigation of the relationship between REV-ERB- α and the kynurenine pathway. While we acknowledge the importance of behavioral and cognitive assessments, our primary objective was elucidating novel mechanistic interactions between these two regulatory components. To this end, we employed both genetic manipulation and pharmacological approaches to allow for more in-depth analysis. Nonetheless, we recognize this as a limitation of our study and have addressed it in the revised manuscript's "Limitations and Future Perspectives" section. Additionally, we have included references to previous findings reporting behavioral changes following exhaustive training in mice (lines 718-720).

Referee #2

"I would like to compliment the authors for substantially improving the manuscript with a lot of edits that had to be made. The authors have addressed most of my concerns and comments in a sufficient manner. Only the following minor comments remain from my side".

1. "Line 316 'Specifically, the JTK CYCLE algorithm was applied to assess circadian oscillations in gene and protein expression over time (Hughes et al., 2010).' Please replace circadian by diurnal again."

Answer: We agree with your commentary. Therefore, we replaced the term circadian with diurnal (line 312).

2. "Line 350 "The EXH group exhibited a reduced amplitude of REV-ERB- α expression in gastrocnemius samples compared to the CTL group ($p=0.001$; Figure 3i)". I cannot follow the authors here, as they state in the rebuttal letter, that "the analysis did not yield statistically significant rhythmicity values". If no significant rhythmicity was found in the present data, further testing for amplitude (in a rhythmicity context) is not justified. I would advise to refrain from using the word "amplitude" as well. Similarly, I don't think the summary sketch on these findings in the abstract is justified, as it suggests continuous (beyond 24 hours) sampling, which was not done. Instead, the authors could perform an equivalent to the 2-way ANOVA and test for an interaction effect, which could hint towards an amplitude effect if tissues were sampled for longer. If true, this could be discussed.

Answer: We understand your concern. Indeed, statistically significant rhythmicity was not detected in the present data. However, the JTK_CYCLE algorithm was applied to evaluate gene expression patterns over 18 hours. Accordingly, we have revised the manuscript to specify "amplitude over 18 hours" rather than using the term "amplitude" alone. Additionally, the summary sketch in the abstract has been updated to reflect this clarification. Although rhythmicity per se was not significant, we believe the comparative analysis of expression dynamics over time still provides valuable insight. The term "amplitude over 18

hours” is used here in a descriptive, rather than rhythmic, context to denote the range of expression variation observed between groups during the sampling window. This approach allows us to highlight potential differences in temporal expression patterns that may be biologically relevant, even without a fully circadian rhythm.

3. *"In some cases, 'circadian rhythm' has now been changed to 'daily rhythm' instead of as stated in the rebuttal letter' diurnal rhythm'. Please make sure to use 'diurnal rhythm' throughout."*

Answer: We agree with your commentary. Therefore, we have checked and updated the entire manuscript to use “diurnal rhythm” specifically.

4. *"I am sorry to say that I don't think that adding Figure 4 has much additional value, the correlation is not very convincing. And particularly for 4c and d, there is no need to show these data again, as Nr1d1 mRNA is indeed well known to oscillate in a circadian manner, which can be referred to. However, it does not change the fact that it was not confirmable in the present data due to lack of samples. 4b is the exception of having additional value, although such delay between mRNA and protein levels was to be expected based on Robles et al. (2014; 10.1371/journal.pgen.1004047), who showed that "the oscillations of almost half of the cycling liver proteome were delayed by more than six hours with respect to the corresponding, rhythmic mRNA". Overall, I think the new section 3.2.5 can mostly be removed. With my initial comment, I only meant to rebut your initial point "Another complicating factor is that mice have an inverted wake-rest cycle compared to humans, affecting REV-ERB- α 's circadian rhythm." The authors could simply address this by referring to Gutierrez-Monreal et al. (<https://doi.org/10.1002/oby.22826>), who compared peak times of expression of a larger composite of clock genes in mouse skeletal muscle and human skeletal muscle, and found for Reverb- α as well as other clock genes that they show mostly consistent peak times, when normalizing the time points to the respective active/rest phase."*

Answer: We agree with your commentary and have revised Figure 4 accordingly. The updated Figure 4a presents the combined temporal profiles of total and phosphorylated REV-ERB- α protein levels alongside *Nr1d1* mRNA expression. Figure 4b illustrates the correlation among these three parameters at baseline

through a heatmap. Additionally, we removed the previous graphs displaying *Nr1d1* oscillation patterns in humans and mice. Instead, we cited a reference highlighting the known differences in REV-ERB- α circadian rhythms between the two species.

5. *"Line 464-466 and Figure 6r and s: Again, I don't think misuse of the term 'amplitude', if meant as now illustrated in the abstract figure with the smaller oscillating wave".*

Answer: We understand your concern. Accordingly, we have updated Figures 6r and 6s to indicate "amplitude over 18 hours" and revised the graphical abstract to reflect this clarification.

6. *"Line 487: It says Figure 4e, but should say Figure 7e."*

Answer: We agree with your comment and have corrected the text accordingly. The reference now correctly cites Figure 7e instead of Figure 4e.

7. *"Line 626-630: The authors could reconsider the relevance of this statement in light of the above mentioned Robles et al. paper. In short, there is a transcript to protein translation delay (for most proteins about 4-6 hours), this is likely to explain the inverse relationship in these data."*

Answer: We agree with your comment and have revised the statement accordingly to reflect the idea suggested by the reviewer (lines 615-619).

Dear Dr da Rocha,

Re: JP-RP-2025-288290R2 "EXHAUSTIVE EXERCISE ABOLISHES REV-ERB- α CIRCADIAN RHYTHM AND SHIFTS THE KYNURENINE PATHWAY TO A NEUROTOXIC PROFILE IN MICE" by Alisson Luiz da Rocha, Ana Paula Pinto, Ivo Vieira de Sousa Neto, Vitor Rosetto Munoz, Bruno Brieda Marafon, Lilian Esleine Costa Mendes da Silva, José Rodrigo Pauli, Dennys Esper Cintra, Eduardo R Ropelle, Fernando Simabuco, Leandro P de Moura, Ellen Cristini de Freitas, Donato Americo Rivas, and Adelino Sanchez Ramos da Silva Silva

We are pleased to tell you that your paper has been accepted for publication in The Journal of Physiology.

Yours sincerely,

Karyn Hamilton
Senior Editor
The Journal of Physiology

If you would like to receive our 'Research Roundup', a monthly newsletter highlighting the cutting-edge research published in The Physiological Society's family of journals (The Journal of Physiology, Experimental Physiology, Physiological Reports, The Journal of Nutritional Physiology and The Journal of Precision Medicine: Health and Disease), please click this link, fill in your name and email address and select 'Research Roundup':
<https://www.physoc.org/journals-and-media/membernews>

- You can help your research get the attention it deserves! Check out Wiley's free Promotion Guide for best-practice recommendations for promoting your work at: www.wileyauthors.com/eeo/guide. You can learn more about Wiley Editing Services which offers professional video, design, and writing services to create shareable video abstracts, infographics, conference posters, lay summaries, and research news stories for your research at: www.wileyauthors.com/eeo/promotion.

EDITOR COMMENTS

Reviewing Editor:

We would like to thank you for your further diligent revisions to your manuscript. Congratulations on a fantastic piece of

work. I look forward to following the future work stemming from this project.

Senior Editor:

Thank you for your careful revisions. We are pleased to accept your manuscript for publication in The Journal of Physiology. Congratulations and thank you for your interest in The Journal!

REFEREE COMMENTS

Referee #1:

I believe the authors have addressed concerns raised--it is an opportunity missed that hippocampal tasks are not tested on the animals, but I recognise that it is too late to address this.

Referee #2:

I would like to thank the authors for all their hard work on revising the manuscript, all my comments have now been fully addressed.